# S4S: Solving for a Fast Diffusion Model Solver

Eric Frankel [1]   Sitan Chen [2]   Jerry Li [1]   Pang Wei Koh [1 3]   Lillian J. Ratliff [1]   Sewoong Oh [1]

## Abstract

Diffusion models (DMs) create samples from a data distribution by starting from random noise and iteratively solving a reverse-time ordinary differential equation (ODE). Because each step in the iterative solution requires an expensive neural function evaluation (NFE), there has been significant interest in approximately solving these diffusion ODEs with only a few NFEs without modifying the underlying model. However, in the few NFE regime, we observe that tracking the true ODE evolution is fundamentally impossible using traditional ODE solvers. In this work, we propose a new method that learns a good solver for the DM, which we call **S**olving **for** the **S**olver (**S4S**). S4S directly optimizes a solver to obtain good generation quality by learning to match the output of a strong teacher solver. We evaluate S4S on six different pre-trained DMs, including pixel-space and latent-space DMs for both conditional and unconditional sampling. In all settings, S4S uniformly improves the sample quality relative to traditional ODE solvers. Moreover, our method is lightweight, data-free, and can be plugged in black-box on top of any discretization schedule or architecture to improve performance. Building on top of this, we also propose **S4S-Alt**, which optimizes both the solver and the discretization schedule. By exploiting the full design space of DM solvers, with 5 NFEs, we achieve an FID of 3.73 on CIFAR10 and 13.26 on MS-COCO, representing a $1.5\times$ improvement over previous training-free ODE methods.

[1]University of Washington [2]Harvard University [3]Allen Institute for AI. Correspondence to: Eric Frankel <ericsf@cs.washington.edu>.

*Proceedings of the $42^{nd}$ International Conference on Machine Learning*, Vancouver, Canada. PMLR 267, 2025. Copyright 2025 by the author(s).

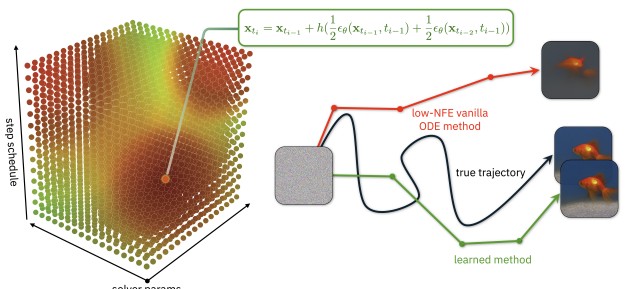

*Figure 1.* Abstraction of the S4S approach. In low-NFE environments, common off-the-shelf ODE solvers are unable to approximate the true diffusion ODE trajectory and produce low-quality samples. In S4S, we learn an optimal combination of solver coefficients and discretization steps that more closely match the *output* of the true ODE trajectory.

## 1. Introduction

Diffusion models (DMs) are a class of powerful models that have revolutionized generative modeling and achieve state-of-the-art performance in a wide number of domains (Sohl-Dickstein et al., 2015; Ho et al., 2020; Song et al., 2021b). Abstractly, DMs learn a *score network* that approximates the time-dependent score function of a diffusion process (Song et al., 2021b; Chen et al., 2023). Sampling from them often involves solving an ordinary differential equation (ODE) referred to as the *diffusion ODE*, where the dynamics are determined by the score network (Song et al., 2021b;a). This ODE typically requires a large number of neural function evaluations (NFEs) to numerically solve, and consequently is quite slow (Ho et al., 2020; Karras et al., 2022). This is directly at odds with many exciting applications of DMs for which low-latency inference is essential, such as robotics (Chi et al., 2024) or game engines (Valevski et al., 2024). Therefore, there is a tremendous amount of interest in understanding how the number of NFEs may be reduced without sacrificing performance.

Methods for enabling DMs to use fewer NFEs generally fall under one of two categories: learning an entirely new model that distills multiple score network evaluations into a single step (**training-based**), or designing efficient diffusion ODE samplers while keeping the score network unchanged (**training-free**). From a practical standpoint, training-based

methods, such as progressive distillation (Salimans & Ho, 2022; Meng et al., 2023) and consistency models (Song et al., 2023) require access to original data samples and substantial computational resources, which may not be available or feasible. Additionally, training-based methods often optimize objectives that fundamentally alter the model's interpretation as a score function, making them unsuitable for tasks that rely on score-based modeling, such as guided generation (Ho & Salimans, 2021), composition (Du et al., 2023), and inverse problem solving (Xu et al., 2024).

For these reasons, we focus on training-free approaches, which requires selecting a discretization of the diffusion ODE and determining both the optimal evaluation time-steps and synthesis strategy to accurately approximate the continuous trajectory. The majority of the literature has focused on choosing a good time-step schedule in this *low NFE regime*—i.e., choosing when to spend our budget of NFEs (Watson et al., 2021; Sabour et al., 2024; Tong et al., 2024; Xue et al., 2024; Chen et al., 2024). Yet, in practice, it is equally important to choose a good solver—this corresponds roughly to choosing how to synthesize these different function evaluations. Most works still rely on traditional "textbook" ODE solvers such as single-step (SS) (Lu et al., 2022a;b) or linear multi-step (LMS) methods (Lu et al., 2022b; Zhang & Chen, 2023). While there is some literature that explores going beyond these solvers (Zheng et al., 2023; Zhang et al., 2023; Zhou et al., 2024a), these approaches only explore narrow components of the sampler design space.

At their heart, off-the-shelf solvers (and much of the prior work on optimizing samplers) seek to approximate the path of the true ODE in discrete time, which can be done given a sufficiently fine discretization (i.e. many NFEs). These methods are carefully crafted so that each step yields an accurate low-degree Taylor approximation of the ODE solution over a small time window. Our key observation is that in the low NFE regime, *this is the wrong thing to target!*, as analytic tools such as low-degree approximation simply do not make sense in the setting where the step-size is gigantic.

We propose to abandon this formalism, and rather to directly optimize a solver to improve performance of the diffusion model. A similar observation was made independently by Shaul et al. (2024b); however, among other issues, the method they derived seeks to completely generalize all previously known solvers. As a result, their solver incorporates large amounts of irrelevant information and optimizes a very complex objective, and is thus unable to match state-of-the-art performance in many settings. In contrast, we give a cleaner, more direct approach for obtaining an optimized solver and demonstrate that our method uniformly improves upon traditional solver performance in virtually all settings we tested.

## 1.1. Contributions

Below we summarize the contributions, breaking them into a description of our efforts on developing a novel method for learning the solver, and a novel method for alternating between learning solver coefficients and the time discretization.

### 1.1.1. SOLVING FOR THE SOLVER (S4S)

The first contribution is a new method for finding numerical solvers for DMs in the low NFE regime. Rather than using any fixed set of pre-existing methods, we instead take the approach of *learning* a good solver for the diffusion model. We call our approach **S**olving **For** the **S**olver, or **S4S**.

Crucially, we seek to find a solver that is good at approximating the overall diffusion process, rather than attempting to discretize any ODE. Indeed, as we demonstrate in Appendix H.2.4, any attempts at maintaining the "standard" invariants that guarantee that traditional solvers track the continuous-time ODE trajectory actively hurt performance. This reinforces our intuition that we must break from this standard approach to obtain the best results.

In somewhat more detail, S4S uses a distillation-style objective for learning solver coefficients. Here, a base "teacher" ODE solver that takes small step sizes—and thus requires many NFEs—provides trajectories that give high sample quality. In turn, a "student" solver with learnable coefficients, given the same noise latent, learns to produce equivalent images with a smaller number of steps. We explain our method in more detail in Section 3.1. Our method has the following advantageous properties.

**Universal improved performance.** In our experiments, we demonstrate that in every setting we tried, our method *universally* improves the FID achieved compared to previous state-of-the-art solvers.

**Plug-in, black-box improvement.** Relatedly, our method can easily be plugged-in in a black-box manner on top of any discretization schedule, and for any architecture. Notably, the gains we achieve from optimizing the solver are *orthogonal* to the gains from optimizing these other axes, e.g. even with a carefully optimized discretization schedule, plugging in S4S will achieve a noticeable improvement in FID. Thus, our method offers a simple way for any practitioner to instantly improve the performance of their generative model.

**Lightweight and data-free.** Our method is lightweight, with minimal computational expense which is comparable to (and often less than) alternative methods for optimizing aspects of the solver, often taking less than an hour on a single A100. Our method is also completely data-free, thus coming at no additional statistical cost to the user.

### 1.1.2. SOLVING FOR THE FULL SAMPLER: S4S-ALT

While S4S by itself already presents uniform and substantial improvements across the board, we find that much of the power of S4S is truly revealed when it is effectively combined with methods for choosing a good discretization. By doing so, we are able to fully exploit the design space of the ODE sampler, something which appears to have been poorly explored in the literature previously. We propose an alternating minimization-based approach that iteratively updates either the coefficients or the discretization schedule one at a time. We call this approach **S4S-Alt**.

While S4S already improves upon previous baselines, by using S4S-Alt to jointly optimize the discretization schedule as well as the solver, we are able to dramatically improve upon state-of-the methods across the board, often by a factor of $1.5 - 2$ or more (see e.g., Table 3 and the tables in the appendix). For example, with only five NFEs, we achieve FID scores of 3.89 on AFHQ-v2, 3.73 on CIFAR-10, 6.25 on FFHQ, 4.39 on class-conditional ImageNet, and 13.26 on MS-COCO with Stable Diffusion. Notably, these numbers are substantially better than what can be achieved by just optimizing the discretization schedule or S4S, separately.

## 2. Background and Related Work

We review background on diffusion models and ODEs, solvers for diffusion ODEs, and learning-based samplers.

### 2.1. Background: Diffusion Models

Let $\mathbf{x}_0 \in \mathbb{R}^d$ be a random variable from an unknown data distribution $p_0(\mathbf{x}_0)$. DMs define a forward process $\{\mathbf{x}_t\}_{t \in [0,T]}$ with $T > 0$ that starts from $\mathbf{x}_0$ and progressively adds Gaussian noise to converge to a marginal distribution, $p_T(\mathbf{x}_T)$, that approximates an isotropic Gaussian, i.e. $p_T(\mathbf{x}_T) \approx \mathcal{N}(\mathbf{x}_T; \mathbf{0}, \tilde{\sigma}^2 \mathbf{I})$ at time $T$ for some $\tilde{\sigma} > 0$ (Ho et al., 2020; Song et al., 2021b). Given $\mathbf{x}_0$, we can characterize the process of adding Gaussian noise by the transition kernel $p_{0t}(\mathbf{x}_t|\mathbf{x}_0) = \mathcal{N}(\mathbf{x}_t; \alpha_t \mathbf{x}_0, \sigma_t^2 \mathbf{I})$, for all $t \in [0,T]$, where $\alpha_t, \sigma_t > 0$ are selected such that the *signal-to-noise ratio* (SNR), $\alpha_t^2/\sigma_t^2$, decays as $t$ increases. Remarkably, Song et al. (2021b) demonstrated that this forward process shares the same marginal distribution $p_t$ as the *probability flow ODE*, a reverse-time ODE starting at $\mathbf{x}_T \sim p_T(\mathbf{x}_T)$ given by

$$d\mathbf{x}_t = \left[ f(t)\mathbf{x}_t - \frac{1}{2}g^2(t)\nabla_{\mathbf{x}} \log p_t(\mathbf{x}_t) \right] dt, \quad (1)$$

where $f(t) = d \log \alpha_t/dt$ and $g(t) = (d\sigma_t^2/dt) - 2(d \log \alpha_t/dt)\sigma_t^2$ (Kingma et al., 2021). Since the score function $\nabla_{\mathbf{x}} \log p_t(\mathbf{x}_t)$ in Eq. (1) is unknown, DMs learn it using a *noise prediction* neural network to minimize

$$\mathcal{L}(\boldsymbol{\theta}) = \mathop{\mathbb{E}}_{\mathbf{x}_0, \boldsymbol{\epsilon}, t}[w(t)\|\boldsymbol{\epsilon}_{\boldsymbol{\theta}}(\mathbf{x}_t, t) - \boldsymbol{\epsilon}\|_2^2]$$

where $\mathbf{x}_0 \sim p_{(}\mathbf{x}_0)$, $\boldsymbol{\epsilon} \sim \mathcal{N}(\mathbf{0}, \mathbf{I})$, $t \sim \mathcal{U}[0, T]$, $w(t)$ is a time-dependent weighting function, and $\mathbf{x}_t = \alpha_t \mathbf{x}_0 + \sigma_t \boldsymbol{\epsilon}$ is a noisy sample at time $t$ (Ho et al., 2020; Lu et al., 2022a). By Tweedie's formula, $\boldsymbol{\epsilon}_{\boldsymbol{\theta}}(\mathbf{x}_t, t)$ learns to approximate $-\sigma_t \nabla_{\mathbf{x}} \log p_t(x)$, thereby defining the *diffusion ODE*

$$d\mathbf{x}_t = \left[ f(t)\mathbf{x}_t + \frac{g^2(t)}{2\sigma_t} \boldsymbol{\epsilon}_{\boldsymbol{\theta}}(\mathbf{x}_t, t) \right] dt, \quad (2)$$

with initial condition $\mathbf{x}_T \sim p_T(\mathbf{x}_T)$. To exactly solve the diffusion ODE at $\mathbf{x}_t$ given an initial value $\mathbf{x}_s$, where $t < s$, Lu et al. (2022a) reparametrizes Eq. (2) in terms of the log signal-to-noise ratio $\lambda_t := \log(\alpha_t/\sigma_t)$, yielding

$$\mathbf{x}_{t_i} = \frac{\alpha_{t_i}}{\alpha_{t_{i-1}}} \mathbf{x}_{t_{i-1}} - \alpha_{t_i} \int_{\lambda_{t_{i-1}}}^{\lambda_{t_i}} e^{-\lambda} \hat{\boldsymbol{\epsilon}}_{\boldsymbol{\theta}}(\hat{\mathbf{x}}_\lambda, \lambda) d\lambda, \quad (3)$$

where $\hat{\mathbf{x}}_\lambda$ and $\boldsymbol{\epsilon}_{\boldsymbol{\theta}}(\hat{\mathbf{x}}_\lambda, \lambda)$ denote the reparametrized forms of $\mathbf{x}_t$ and $\boldsymbol{\epsilon}_{\boldsymbol{\theta}}(\mathbf{x}_t, t)$ in the $\lambda$ domain.

### 2.2. Background: Solving the Diffusion ODE

Sampling from a DM requires numerically solving the diffusion ODE in Eq. (2). Given a decreasing sequence of $N$ discretization steps $\{t_i\}_{i=0}^N$ from $t_0 = T$ to $t_N = 0$, we iteratively compute a sequence of estimates $\{\tilde{\mathbf{x}}_{t_i}\}_{i=0}^N$ starting from $\tilde{\mathbf{x}}_{t_0} = \mathbf{x}_T \sim \mathcal{N}(\mathbf{x}_T; \mathbf{0}, \tilde{\sigma}^2 \mathbf{I})$ such that the *global truncation error* between $\tilde{\mathbf{x}}_{t_N}$ and the true solution $\mathbf{x}_{t_N}$ is low. The standard approach of controlling this error is to bound the *local truncation error* between $\tilde{\mathbf{x}}_{t_i}$ and $\mathbf{x}_{t_i}$ at each $t_i$. Since Eq. (3) gives the exact solution of the diffusion ODE given an initial value $\tilde{\mathbf{x}}_{t_{i-1}}$, an accurate approximation of the integral in turn provides an accurate approximation $\tilde{\mathbf{x}}_{t_i}$ for the true solution at time $t_{i-1}$. One can take a Taylor expansion of $\hat{\boldsymbol{\epsilon}}_{\boldsymbol{\theta}}(\hat{\mathbf{x}}_\lambda, \lambda)$ about $\lambda_{t_{i-1}}$ in Eq. (3), yielding

$$\tilde{\mathbf{x}}_{t_i} = \frac{\alpha_{t_i}}{\alpha_{t_{i-1}}} \tilde{\mathbf{x}}_{t_{i-1}} - \alpha_{t_i} \sum_{n=0}^{k-1} \hat{\boldsymbol{\epsilon}}_{\boldsymbol{\theta}}^{(n)}(\hat{\mathbf{x}}_{\lambda_{t_{i-1}}}, \lambda_{t_{i-1}})\psi_n(h)$$
$$+ \mathcal{O}(h_i^{k+1}), \quad (4)$$

for some $\psi_n(h)$ depending on $n$, $\lambda_{t_i}$, and $\lambda_{t_{i-1}}$; see Appendix B.1 for further details. Computing such $k$-th order approximation requires accurate estimates of the derivatives $\hat{\boldsymbol{\epsilon}}_{\boldsymbol{\theta}}^{(n)}$ up to order $n = k - 1$. Existing methods use two main approaches from ODE literature: single-step methods (Lu et al., 2022a;b; Zheng et al., 2023; Zhao et al., 2023; Zhang & Chen, 2023; Karras et al., 2022), which use $k-1$ intermediate points in $(t_i, t_{i-1})$, and linear multi-step methods (Lu et al., 2022b; Zheng et al., 2023; Zhao et al., 2023; Zhang & Chen, 2023; Liu et al., 2022), which use information from $k - 1$ previous steps. For low order methods ($k \leq 4$), under appropriate regularity conditions (see Appendix B.2) and when $h_{max} := \max_{1 \leq i \leq N} h_i$ is bounded by $\mathcal{O}(1/N)$, these methods achieve local truncation error of $\mathcal{O}(h_i^{k+1})$ and therefore global error of $\mathcal{O}(h_{max}^k)$.

When the number of NFEs is large and thus $h_{max}$ is small, local truncation error control yields high quality samples (Lu et al., 2022a;b; Zhang & Chen, 2023). However, with few NFEs and large $h_{max}$, the higher-order Taylor errors dominate, leading to large global error. In contrast, our approach in Eq. (6) directly minimizes the global error.

## 2.3. Related Work: Learned Samplers

In practice, no single pair of ODE solver and a time discretization generates high quality samples universally across various datasets and model architectures, e.g. Appendix H.4 and Tong et al. (2024). This inspired *learning*-based methods for deriving ODE solvers and time discretizations adapted to the given task and architecture. We give a brief survey here and discuss in detail in Appendix A. One popular approach exclusively learns the discretization steps (Watson et al., 2021; Sabour et al., 2024; Xue et al., 2024; Tong et al., 2024; Chen et al., 2024). Our approach S4S learns the solver coefficients, complementing the gains of such methods and universally improving the performance in all scenarios, as seen in Table 2 and comprehensively in Appendix H.4. Another line of research focuses on optimizing only the solver coefficients (Zheng et al., 2023; Zhang et al., 2023), or jointly optimizing both solver coefficients and time discretizations (Zhou et al., 2024a; Zheng et al., 2023; Liu et al., 2023; Shaul et al., 2024a). However, these methods are designed to minimize the *local* approximation error through the same methods as in Eq. (4) or by closely matching the *entire trajectory* of the teacher solver. Instead, by minimizing the *global* error by matching the *end* of the teacher trajectory, as in Eq. (6), S4S significantly improves over these approaches. Closest to our approach is BNS (Shaul et al., 2024b), which learns both the solver coefficients and time discretizations to minimize global error. We provide comparisons in Table 4 and explain our improvements over BNS in Appendix A.3.

## 3. Learning Diffusion Model Samplers

We detail our strategy for creating DM samplers that produce high-quality samples using a small number of NFEs. We exploit the full design space of diffusion model solvers by learning both the coefficients and discretization steps of the sampler, as both necessarily interact with one another. We first characterize this design space by providing a general formulation for three general types of diffusion ODE solvers: single-step (SS), linear multi-step (LMS), and predictor-corrector methods (PC). We then describe the objective we minimize to directly control the global error. Next, given a pre-specified set of discretization steps, we introduce our algorithm for learning only the solver coefficients; this uniformly improves performance over hand-crafted solvers for an equivalent number of NFEs. Finally,

we describe our method for learning *both* the solver coefficients *and* the discretization steps.

## 3.1. S4S: Learning Solver Coefficients

For a learned score network and initial noise latent $\mathbf{x}_T \sim \mathcal{N}(\mathbf{0}, \tilde{\sigma}^2 \mathbf{I})$, one can sample from diffusion ODE using an appropriate sequence of pre-determined discretization steps $\{t_i\}_{i=0}^N$ and an ODE solver $\Psi$ determined by its coefficients $\phi$ and the number of steps $k$ it uses. For SS and LMS solvers, we write their estimate of the next step as

$$\tilde{\mathbf{x}}_{t_i} = \frac{\alpha_{t_i}}{\alpha_{t_{i-1}}} \tilde{\mathbf{x}}_{t_{i-1}} - \sigma_{t_i}(e^{h_i} - 1)\Delta_i(\phi), \qquad (5)$$

where $\Delta_i(\phi)$ represents the *increment* of the solver as a function of the coefficients $\phi$. We explicitly define $\Delta_i(\phi)$ in Table 1. A PC solver further refines this initial prediction, by subsequently applying Eq. (5) again with new coefficients. We provide the intuition behind this formulation in Appendix B.1 and equivalent examples for a data prediction model in Appendix C. To denote the fact that a learned solver uses $k$ steps of information, we abuse notation and refer to it as having order $k$.

We propose **S**olving **for** the **S**olver (**S4S**) in Algorithm 1 to learn these coefficients to adapt to the problem instance of the given score network. Consider the outputs from a "teacher" solver, $\Psi^*(\mathbf{x}_T)$, which accurately solves the diffusion ODE. We aim to minimize the *global error* between the sample $\Psi_\phi(\mathbf{x}_T)$ generated by sequentially applying $\Psi_\phi$ from $t_0 = T$ to $t_N = 0$ and the sample from the teacher:

$$\mathcal{L}(\phi) = \min_{\phi} \mathbb{E}_{\mathbf{x}_T \sim \mathcal{N}(\mathbf{0}, \tilde{\sigma}^2 \mathbf{I})} [d(\Psi_\phi(\mathbf{x}_T), \Psi^*(\mathbf{x}_T))], \quad (6)$$

where $d(\cdot, \cdot)$ is an appropriate distance function that is differentiable, non-negative, and reflexive. For now, $\{t_i\}_{i=0}^N$ is a pre-determined discretization schedule, though we also propose learning the discretizations in Section 3.2. We emphasize the importance of learning a solver with respect to the global error: although some existing works try to match the teacher solver's trajectory, many teacher trajectories contain pathologies that are subsequently distilled into the student; see Appendix B.3 for further discussion. While this method, as stated, already improves performance out-of-the-box, we now detail two optimizations that further improve our performance.

### 3.1.1. TIME-DEPENDENT COEFFICIENTS

Traditional methods for solving ODEs (e.g. Adams-Bashforth or Runge-Kutta) are often defined by a constant set of coefficients, regardless of what time step along the ODE they are estimating. While this is not uniformly the case for diffusion ODE solvers, many keep coefficients fixed across steps of solving the reverse-process; see Appendix C.2. This fails to fully capture the complexity of

| Solver Type | $\Delta_i(\phi)$ | $\phi$ | NFEs per Step | # Params. |
|---|---|---|---|---|
| LMS | $\sum_{j=1}^{k} b_{j,i} \boldsymbol{\epsilon}_{\boldsymbol{\theta}}(\tilde{\mathbf{x}}_{t_{i-j}}, t_{i-j})$ | $\{b_{j,i}\}$ | 1 | $k(2N+1-k)/2$ |
| SS | $\sum_{j=1}^{k} b_{j,i}\kappa_j, \ \kappa_j = \boldsymbol{\epsilon}_{\boldsymbol{\theta}}\left(\tilde{\mathbf{x}}_{t_{i-1}} + \sum_{l=1}^{j-1} a_{j,i,l}\kappa_l, t_{i-1} + c_{j,i}\right)$ | $\{b_{j,i}, a_{j,i,l}, c_{j,i}\}$ | $k$ | $(k^2 + k - 1)N$ |
| LMS + PC | $\sum_{j=1}^{k} a_{j,i}^c \boldsymbol{\epsilon}_{\boldsymbol{\theta}}(\tilde{\mathbf{x}}_{t_{i-j}}, t_{i-j})$ | $\{b_{j,i}\} + \{a_{j,i}^c\}$ | 1 | $k(2N+1-k)$ |

*Table 1.* We apply S4S to three types of diffusion ODE solvers; we show their increment ($\Delta_i$), learnable parameters, number of NFEs per step, and total parameter count over $N+1$ steps. By default, we use a linear multi-step predictor for the PC method, so $\{a_{j,i}^c\}$ refer to coefficients during the correction step, and the total set of learnable parameters accounts for the underlying multi-step predictor.

---

**Algorithm 1** S4S

**Require:** Coefficient parameters $\phi$, student solver $\Psi_\phi$, teacher solver $\Psi^*$, distance metric $d$, and $r$.
1: $\mathcal{D} \leftarrow \{(\mathbf{x}_T', \mathbf{x}_T, \Psi^*(\mathbf{x}_T)) \mid \mathbf{x}_T \sim \mathcal{N}(\mathbf{0}, \tilde{\sigma}^2 \mathbf{I}), \mathbf{x}_T' = \mathbf{x}_T\}$ // *Generate data $\mathcal{D}$*
2: **while** not converged **do**
3:     $(\mathbf{x}_T', \mathbf{x}_T, \Psi^*(\mathbf{x}_T)) \sim \mathcal{D}$
4:     $\mathcal{L}(\phi, \mathbf{x}_T') = d(\Psi_\phi(\mathbf{x}_T'), \Psi^*(\mathbf{x}_T))$ subject to $\mathbf{x}_T' \in B(\mathbf{x}_T, r\sigma_T)$
5:     Update $\phi$ and $\mathbf{x}_T'$ using the corresponding gradients $\nabla\mathcal{L}(\phi, \mathbf{x}_T')$
6:     $\mathbf{x}_T' \leftarrow \mathbf{x}_T + \mathbf{1}[\|\mathbf{x}_T' - \mathbf{x}_T\|_2 > r] \cdot r \frac{\mathbf{x}_T' - \mathbf{x}_T}{\|\mathbf{x}_T' - \mathbf{x}_T\|_2}$ // *Projected SGD*
7:     Update $\mathcal{D}$ with the new $\mathbf{x}_T'$
8: **end while**

---

diffusion ODEs: the score network increasingly suffers from prediction error as the marginal distribution $p_t(\mathbf{x}_t)$ resembles Gaussian noise less and less, while estimation error that occurs at a noisy time step propagates through the estimated trajectory differently than at a "cleaner" step. Accordingly, as an additional optimization, S4S learns *time-dependent* coefficients, as exemplified by the dependence on the current iteration $i$ in Table 1. We ablate the design decision to use time-dependent coefficients in Appendix H.2; time-dependent coefficients significantly outperform the use of fixed coefficients.

### 3.1.2. RELAXED OBJECTIVE

For each student solver $\Psi_\phi$, the number of both NFEs and learnable parameters is determined by the type of solver, the number of discretization steps, and the step parameter $k$ of the solver, as displayed in Table 1. Accordingly, when the target solver uses few NFEs, the number of learnable parameters may be very low, e.g. 6 parameters for LMS when $N = k = 3$. This can make optimizing Eq. (6) difficult: indeed, given an initial condition $\mathbf{x}_T$, our objective tries to ensure that $\Psi_\phi(\mathbf{x}_T) = \Psi^*(\mathbf{x}_T)$. Given the small number of learnable parameters, however, the student solver will almost always produce an output with non-trivial truncation error. As a result, though our learned coefficients may be successful at reducing the global error, they might

nonetheless underfit the objective and fail to fully achieve the expected performance improvements.

Instead, similar to Tong et al. (2024), we propose a relaxation of our training objective that is easier to optimize with a limited number of parameters. In particular, rather than forcing the student solver to exactly reproduce the teacher's output for $\mathbf{x}_T$, we instead only require the existence of an input $\mathbf{x}_T'$ sufficiently close to $\mathbf{x}_T$ (i.e. within a bounded radius) such that $\Psi_\phi(\mathbf{x}_T') = \Psi^*(\mathbf{x}_T)$. As a result, so long as $\mathbf{x}_T'$ is appropriately close to $\mathbf{x}_T$, the average global error of the learned student model can still be quite low, while mitigating the difficulty of the objective. Concretely, our relaxed objective is expressed as

$$\mathcal{L}_{\text{relax}}(\phi) = \min_{\phi} \ \mathbb{E}_{\mathbf{x}_T \sim \mathcal{N}(0, \sigma_T^2 \mathbf{I})} [J(\mathbf{x}_T, \mathbf{x}_T')]$$
$$J(\mathbf{x}_T, \mathbf{x}_T') = \min_{\mathbf{x}_T' \in B_r(\mathbf{x}_T)} d(\Psi_\phi(\mathbf{x}_T'), \Psi^*(\mathbf{x}_T)) \quad (7)$$

where $B_r(\mathbf{x}) := \{\mathbf{x}' \mid \|\mathbf{x} - \mathbf{x}'\|_2 \leq r\tilde{\sigma}\}$ is the $L_2$ ball of radius $r\tilde{\sigma}$ about $\mathbf{x}$. This objective has several appealing properties. First, in Appendix D.2, we empirically verify, similar to Tong et al. (2024), that this objective is easier to solve than our original objective, which we recover when $r = 0$. Moreover, under appropriate assumptions on the solver, we can ensure that distribution generated by the learned solver, $p_\phi(\mathbf{x}_0)$, and that of the teacher solver, $p^*(\mathbf{x}_0)$, are sufficient close; see Appendix D.1 for details. Finally, although we

minimize this objective during training, at inference time, we only use the initial condition $\mathbf{x}_T \sim p_T(\mathbf{x}_T)$ rather than finding and using $\mathbf{x}'_T \sim B_r(\mathbf{x}_T)$ as an initial condition.

## 3.2. S4S-Alt: Coefficients *and* Time Steps

While learning the solver coefficients alone improves the quality of samples, the choice of discretization steps remains crucial for achieving optimal performance. In that vein, we present **S4S-Alt**, which learns both solver coefficients and discretization steps by using alternating minimization over objectives for the coefficients or the discretization steps.

### 3.2.1. DISCRETIZATION STEP PARAMETRIZATION

When sampling from a DM, the choice of discretization steps determines (1) the expected amount of signal-to-noise present in an estimated sample, (2) the error present in the score network's prediction, and (3) the amount of error propagated by using estimated trajectory points as input to the score network. We take these consequences into account when parametrizing a learned set of discretization steps by separating the learned steps into two parts. First, we use a set of time steps, $\{t_i^{\boldsymbol{\xi}}\}_{i=0}^{N+1}$, that is parametrized by a learnable vector $\boldsymbol{\xi} \in \mathbb{R}^{N+1}$ used for determining the step size and SNR parameters, thereby accounting for (1). We explicitly parameterize $t_i^{\boldsymbol{\xi}}$ such that it is a monotonically decreasing sequence of parameters between 0 and $T$, i.e. $t_0^{\boldsymbol{\xi}} = T > t_1^{\boldsymbol{\xi}} > \cdots > t_N^{\boldsymbol{\xi}} = 0$; see Appendix E.1 for an explicit description of this parametrization. Second, we use a modified set of time steps as input to the score network to mitigate (2) and (3). Specifically, we use a set of decoupled steps $\{t_i^c = t_i^{\boldsymbol{\xi}} + \boldsymbol{\xi}_i^c\}_{i=0}^N$ as input to the score network, where $\boldsymbol{\xi}^c \in \mathbb{R}^{N+1}$; we describe the construction of $\boldsymbol{\xi}^c$ in Appendix E.2. Under this parametrization, the update step of the $k$-step LMS in Eq. (5) and Table 1 is:

$$\tilde{\mathbf{x}}_{t_i^{\boldsymbol{\xi}}} = \frac{\alpha_{t_i^{\boldsymbol{\xi}}}}{\alpha_{t_{i-1}^{\boldsymbol{\xi}}}}\tilde{\mathbf{x}}_{t_{i-1}} - \sigma_{t_i^{\boldsymbol{\xi}}}(e^{h_i}-1)\sum_{j=0}^{p} b_{j,i}\boldsymbol{\epsilon_\theta}(\tilde{\mathbf{x}}_{t_{i-k+j}}, t_{i-k+j}^c)$$

where $h_i = t_i^{\boldsymbol{\xi}} - t_{i-1}^{\boldsymbol{\xi}}$. For simplicity, we denote the collection of learnable time parameters as $\boldsymbol{\Xi} := \{\boldsymbol{\xi}, \boldsymbol{\xi}^c\}$. Consequently we represent a solver with learnable coefficients *and* time steps as $\Psi_{\boldsymbol{\phi}, \boldsymbol{\Xi}}$ and its outputs as $\Psi_{\boldsymbol{\phi}, \boldsymbol{\Xi}}(\mathbf{x}_T)$.

### 3.2.2. ALTERNATING OPTIMIZATION

We next consider how to optimize both the solver as well as the discretization schedule. We propose an iterative approach, **S4S-Alt**, that alternates between optimizing the time steps and the solver coefficients. Formally, at iteration $k$,

we solve the objectives

$$\boldsymbol{\Xi}_k = \arg\min_{\boldsymbol{\Xi}} \mathbb{E}_{\mathbf{x}_T \sim \mathcal{N}(\mathbf{0}, \tilde{\sigma}^2 \mathbf{I})}[d(\Psi_{\boldsymbol{\phi}_{k-1}, \boldsymbol{\Xi}_{k-1}}(\mathbf{x}_T), \Psi^*(\mathbf{x}_T))],$$

$$\boldsymbol{\phi}_k = \arg\min_{\boldsymbol{\phi}} \mathbb{E}_{\mathbf{x}_T \sim \mathcal{N}(\mathbf{0}, \tilde{\sigma}^2 \mathbf{I})}[d(\Psi_{\boldsymbol{\phi}_{k-1}, \boldsymbol{\Xi}_k}(\mathbf{x}_T), \Psi^*(\mathbf{x}_T))].$$
(8)

In the first objective, we learn only $\boldsymbol{\Xi}_k$ using the LD3 objective (Tong et al., 2024) from a student solver with coefficients and time steps initialized at $\boldsymbol{\phi}_{k-1}$ and $\boldsymbol{\Xi}_{k-1}$, respectively. In the second, we learn $\boldsymbol{\phi}_k$ from a solver initialized at the newly learned time steps $\boldsymbol{\Xi}_{k-1}$ and coefficients $\boldsymbol{\phi}_{k-1}$.

A natural alternative to this approach would be to optimize the coefficients and time steps simultaneously. However, in our experiments, we found that optimizing both simultaneously presents several challenges, namely that the optimization landscape becomes significantly more complex due to the interaction between the solver coefficients and time steps. Additionally, we found that learning both jointly has a greater risk of over-fitting. We found that S4S-Alt performed significantly better in practice, as seen in Table 6.

## 3.3. Implementation Details

Below, we discuss the practical details used for S4S. For ease of notation, we first ground our explanation in the version of S4S that only learns coefficients before discussing details specific to our S4S-Alt. We direct explicit queries about hyperparameters, etc. to Appendix G.2.

**Practical Objective.** Despite formulating our relaxed objective in Eq. (7), optimizing it in practice is still unclear. To do so, we treat our optimization problem as jointly optimizing both $\boldsymbol{\phi}$ and $\mathbf{x}'_T$, using projected SGD to enforce the constraint that $\mathbf{x}'_T$ remain close to $\mathbf{x}_T$. Concretely, this is

$$\mathcal{L}_{\text{relax}}(\boldsymbol{\phi}, \mathbf{x}'_T) := \mathbb{E}_{\mathbf{x}_T \sim \mathcal{N}(\mathbf{0}, \tilde{\sigma}^2 \mathbf{I})}[d(\Psi_{\boldsymbol{\phi}}(\mathbf{x}'_T), \Psi^*(\mathbf{x}_T))],$$
$$\text{subj. to } \mathbf{x}'_T \in B_r(\mathbf{x}_T). \quad (9)$$

In practice, we use LPIPS as our distance metric, a common loss for distillation-based methods (Salimans & Ho, 2022; Song et al., 2023); for other modalities, alternatively appropriate distance metrics should be used. We ablate the decision to use LPIPS in Section 4.2.

**Algorithm Details.** The algorithm for S4S learning coefficients is displayed in Algorithm 1. First, we collect a dataset from a sequence of noise latents used to create samples from the teacher solver $\Psi^*(\mathbf{x}_T)$. Initially, we use the same initial condition for both the student and teacher solver, i.e. $\mathbf{x}'_T = \mathbf{x}_T$. At each iteration, for a given batch, we compute the loss between the output of our learned solver $\Psi_{\boldsymbol{\phi}}(\mathbf{x}'_T)$ and $\Psi^*(\mathbf{x}_T)$, and use backpropagation to get the gradients of this loss with respect to $\boldsymbol{\phi}$ and $\mathbf{x}'_T$. To enforce our constraint on $\mathbf{x}'_T$, we use projected SGD to ensure it remains

inside of $B_r(\mathbf{x}_T)$; for coefficients, we can use an arbitrary method for applying the gradients, although momentum-based methods work best empirically. Notably, after we update $\mathbf{x}'_T$, we keep it with its original $(\mathbf{x}_T, \Psi^*(\mathbf{x}_T))$ pair, and update the dataset with the new noise latent. We also optimize our computation of the gradient computation graph; see Appendix F.2 for more details.

**Initialization.** A natural question to consider is how the student ODE solver coefficients may be initialized. Since our approach generally subsumes common diffusion ODE solvers, including the best-performing methods like DPM-Solver++ (Lu et al., 2022b), iPNDM (Zhang & Chen, 2023), and UniPC (Zhao et al., 2023), we can initialize $\phi$ with the same coefficients as these methods. This can be interpreted as wrapping one of these traditional solvers in our lightweight approach; in this setting where just coefficients are learned, we refer to this as e.g. iPNDM-S4S. Alternatively, we could consider initializing the coefficients according to a Gaussian. We ablate this decision in Appendix H.2.1, finding that solver initialization outperforms Gaussian initialization.

**Algorithms for Learning Coefficients and Time Steps.** In practice, when learning both time steps and solver coefficients for a student solver $\Psi_{\phi, \Xi}$, S4S optimizes an equivalent, alternating version of Eq. (9) (and equivalently for jointly learning coefficients); likewise, the pseudocode for doing so is quite similar, which we detail in Appendix F.1. Nonetheless, in practice, learning $\Psi_{\phi, \Xi}$ generally requires a larger dataset compared to just learning the coefficients, largely attributable to a larger number of parameters. We ablate performance with dataset size in Appendix H.2.3.

## 4. Experiments

We evaluate S4S on a number of pre-trained diffusion models trained on common image datasets. We use pixel-space diffusion models for CIFAR-10 (32x32), FFHQ (64x64), and AFHQv2 (64x64), each having an EDM-style backbone (Karras et al., 2022). We also use latent diffusion models, including LSUN-Bedroom (256x256) and class-conditional ImageNet (256x256) with a guidance scale of 2.0. Finally, we present both qualitative and quantitative results for Stable Diffusion v1.4 at 512×512 pixels with a variety of guidance scales. We provide precise experimental details in Appendix G for all sets of experiments, including choice of teacher solver, dataset size, and selection of noise radius $r$. We use the Frechet Inception Distance score (FID) as a metric for image quality on all datasets using 30k samples generated from MS-COCO captions for evaluating Stable Diffusion and 50k samples for all other datasets.

First, we show the benefits of **S4S** as a *standalone* wrap-

| Schedule | Method | NFE=4 | NFE=6 | NFE=8 |
|---|---|---|---|---|
| **CIFAR-10** | | | | |
| EDM | UniPC | 50.63 | 19.47 | 9.68 |
| | UniPC-S4S | **44.30** | **17.80** | **9.05** |
| | iPNDM | 29.50 | 9.75 | 5.24 |
| | iPNDM-S4S | **25.74** | **8.81** | **4.98** |
| | DPM-v3 | 34.39 | 18.44 | 7.39 |
| LD3 | UniPC | 15.83 | 3.55 | 2.87 |
| | UniPC-S4S | **13.46** | **3.17** | **2.67** |
| | iPNDM | 10.93 | 5.40 | 2.75 |
| | iPNDM-S4S | **9.30** | **4.76** | **2.61** |
| | DPM-v3 | 29.86 | 10.69 | 3.59 |
| **ImageNet** | | | | |
| $t$-Unif | UniPC | 53.22 | 10.97 | 5.53 |
| | UniPC-S4S | **45.53** | **10.09** | **5.19** |
| | iPNDM | 36.23 | 16.15 | 7.93 |
| | iPNDM-S4S | **31.81** | **14.85** | **7.53** |
| LD3 | UniPC | 11.33 | 4.74 | 4.87 |
| | UniPC-S4S | **10.56** | **4.54** | **4.58** |
| | iPNDM | 6.45 | 4.70 | 4.91 |
| | iPNDM-S4S | **6.05** | **4.57** | **4.68** |

*Table 2.* FIDs on CIFAR-10 and ImageNet show that common diffusion ODE solvers can be improved by S4S initialized at that solver for almost all NFEs and schedules.

per around learnable third-order multi-step versions of the best current ODE solvers: UniPC (Zhao et al., 2023) and iPNDM (Zhang & Chen, 2023). Here, we initialize our student solver to have the same coefficients as their unlearned counterparts before optimizing our relaxed objective. When possible, we also compare with DPM-Solver-v3 (Zheng et al., 2023), which learns coefficients, but only to attain a guarantee on local truncation error. We evaluate our learned solvers on seven discretization schedule methods, ranging from common heuristics to modern step-selection methods, with further details in Appendix G.1. We also characterize the performance of S4S on learnable single-step methods, which can be found in Appendix H.1.

Next, we evaluate **S4S-Alt** against several methods of *learning* sampler attributes, including AMED-Plugin (Zhou et al., 2024a) and BNS (Shaul et al., 2024b), in sample quality and computational efficiency. We instantiate S4S-Alt as a LMS method initialized with iPNDM coefficients and LD3 discretization; this limits the amount of overfitting to the training data due to fewer parameters relative to SS and PC methods. Finally, we ablate key design decisions in S4S.

### 4.1. Main Results

When used as a wrapper for learning solver coefficients, S4S almost **uniformly** improves image generation quality

| Method | NFE=4 | NFE=6 | NFE=8 |
|---|---|---|---|
| **CIFAR-10** | | | |
| Best DPM-v3 | 17.88 | 7.32 | 3.59 |
| Best Trad. (LD3) | 10.93 | 3.55 | 2.75 |
| Best S4S | 8.25 | 3.17 | 2.61 |
| S4S Alt | **6.35** | **2.67** | **2.39** |
| **MS-COCO** | | | |
| DPM-v3 | 23.90 | 15.22 | 12.10 |
| Best Trad. (LD3) | 20.22 | 12.33 | 11.30 |
| Best S4S | 19.14 | 11.97 | 10.82 |
| S4S Alt | **16.05** | **11.17** | **10.68** |

*Table 3.* S4S-Alt consistently offers significant improvements in FID over best-performing alternatives at a given number of NFEs.

| Method | CIFAR | | MS-COCO | |
|---|---|---|---|---|
| | NFE | FID | NFE | FID |
| S4S-Alt | 7 | 2.52 | 6 | 11.17 |
| S4S | 10 | 2.18 | 8 | 10.84 |
| LD3 | 10 | 2.32 | 8 | 12.28 |
| DPM-v3 | 10 | 2.32 | 8 | 12.10 |
| BNS[†] | 8 | 2.73 | 12 | 20.67 |
| PD[†] | 8 | 2.57 | - | - |
| ECM[†] | 2 | 2.20 | - | - |
| iCT-deep[†] | 1 | 2.51 | - | - |

*Table 4.* Number of NFEs required to match/beat S4S-Alt performance on CIFAR and MS-COCO. † denotes that results were taken from original papers. PD refers to Progressive Distillation (Salimans & Ho, 2022), ECM to Easy Consistency Models (Geng et al., 2025), iCT-deep to Improved Consistency Training (Song & Dhariwal, 2024). Red cells are methods that cannot match S4S-Alt in our experiments w/ our NFE settings or in recorded experiments.

across datasets, solver types, and discretization methods in the few-NFE regime. Our full results are available in Appendix H.4, while we present a selection of results on CIFAR-10 and ImageNet in Table 2. We observe that the size of the improvement that S4S provides is dependent on the underlying discretization schedule and solver type, and while S4S always improves performance for any discretization schedule, the amount of the improvement varies across different choices of schedule. When we both optimize the solver and the schedule, i.e., with S4S-Alt, we obtain *significantly* greater improvements compared to prior state-of-the-art. We display some of these results in Table 3, where we compare against methods that learn a *single* dimension of the sampler: the best "traditional" ODE solver using the learned LD3 discretization schedule, the best FID DPM-Solver-v3 across all schedules, and the best S4S solver across all schedules; see Appendix H.4 for the full set of FID values across our experiments. S4S-Alt achieves extremely strong performance relative to simple learned methods. We

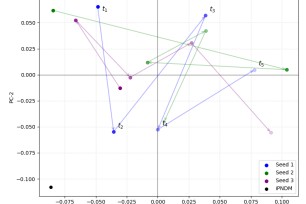
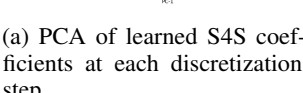
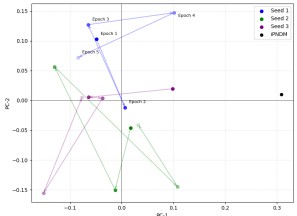

(a) PCA of learned S4S coefficients at each discretization step.

(b) PCA of learned S4S coefficients at each epoch of training.

*Figure 2.* PCA of learned S4S coefficients at a each point of the reverse process or at b each training epoch; darker points refer to earlier values in the reverse process or training. We initialize S4S coefficients at iPNDM and learn a solver with 5 NFEs and order 3. In a, we take the PCA of the combined set of final learned coefficients $\{(b_{1,i}, b_{2,i}, b_{3,i})\}_{i=1}^{5}$ across the three training random seeds used. We also include the iPNDM coefficients in the PCA, using a total of 16 vectors in $\mathbb{R}^3$. In b, we concatenate the learned coefficient vectors at the end of each epoch, resulting in a vector of dimension $\mathbb{R}^{15}$ for each epoch. We again perform PCA on a collection of 16 of these vectors, again including iPNDM as a reference point.

also provide qualitative comparisons in Appendix H.5.

Finally, we provide a detailed comparison of S4S-Alt to methods that learn aspects of the solver, as well as *training-based* distillation methods, in Appendix H.3 and briefly in Table 4. S4S-Alt outperforms the *vast majority* of learnable solver methods and achieves competitive performance to training-based methods for a fraction of the compute.

### 4.2. Ablations

Table 5 shows ablation on the solver order in learned LMS models. In both versions of S4S, excessively large order tends to *decrease* performance; despite setting $r$ proportionally to the larger number of parameters, using information from distant time steps hurts output sample quality. We also characterize the importance of our alternating minimization objective for S4S-Alt. As an alternative, we consider learning both the solver coefficients and discretization steps simultaneously using the same objective; see Appendix H.2.2 for an explicit description. We present our results in Table 6. Finally, we characterize many more design decisions made over our solver; see Appendix H.2.

## 5. Conclusion

We introduce S4S, a method for learning DM solvers in the low NFE regime. Our approach matches the output of a teacher solver while simplifying the optimization landscape for learning a student solver. While we achieve superior results, there are nonetheless limitations and opportunities

| Method | Order | NFE=4 | NFE=6 | NFE=8 |
|--------|-------|-------|-------|-------|
| S4S | 3 | 14.24 | 5.45 | 3.55 |
| | 4 | 13.94 | 5.68 | 3.61 |
| | 6 | - | 6.11 | 3.89 |
| S4S-Alt | 3 | 10.63 | 4.62 | 3.15 |
| | 4 | 10.21 | 4.40 | 3.24 |
| | 6 | - | 4.83 | 3.42 |
| Baseline | 3 | 16.68 | 6.19 | 3.75 |

*Table 5.* Effect of solver order on FID for FFHQ. Both S4S methods are LMS initialized with iPNDM, and standalone S4S uses LD3 schedule. Cells that have *worse* performance than traditional iPNDM with LD3 are highlighted in red. Excessively high order degrades quality in both versions of S4S.

| Method | Order | NFE=4 | NFE=6 | NFE=8 |
|--------|-------|-------|-------|-------|
| S4S-Alt | 3 | 6.35 | 2.67 | 2.39 |
| Joint Obj. | 3 | 6.81 | 3.28 | 2.91 |
| Joint Obj. | Eq-NFE | 6.42 | 3.37 | 3.76 |
| iPNDM-S4S | 3 | 9.30 | 4.76 | 2.61 |
| iPNDM | 3 | 10.93 | 5.40 | 2.75 |

*Table 6.* Using a *joint* objective for learning both coefficients and time steps, and the interaction of the joint objective with the order of the underlying LMS method vs. S4S-Alt on CIFAR-10. Eq-NFE denotes having an order equal to the number of NFEs used, e.g. order 6 at 6 NFEs. Orange indicates worse performance than S4S on iPDNM; red indicates worse than traditional iPNDM.

for future work: 1) we only experimented on ODE solvers, 2) coefficients must be learned for each number of NFEs and cannot be re-used, and 3) we learn dataset-level coefficients rather than sample-level coefficients. Our low-NFE performance also lags behind the very best training-based approaches.

## Impact Statement

In terms of broader impact, the techniques in this work may be used to improve samples produced for beneficial purposes, such as private data synthesis; moreover, the efficient generation that S4S provides helps reduce the energy usage in this era of modern machine learning. Nonetheless, these models could also be used to generate highly realistic images, audio, or videos that could be used for harmful purposes. We believe technical innovations such as ours should be balanced with safety measures.

## Acknowledgments

SO and EF were supported in part by NSF Awards 2019844, 2112471, and 2229876, and Microsoft Grant for Customer Experience Innovation. LJR and EF were supported in part by NSF CPS-1844729, and ONR YIP N00014-20-1-2571. EF is also supported in part by the NSF Graduate Research Fellowship Program. PWK is supported by the Singapore National Research Foundation and the National AI Group in the Singapore Ministry of Digital Development and Information under the AI Visiting Professorship Programme (award number AIVP-2024-001), and by the AI2050 program at Schmidt Sciences. SC was supported in part by NSF IIS-2331831, NSF CCF-2441635, and the Harvard Dean's Competitive Fund for Promising Scholarship.

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

# A. Comparisons with Existing Works

Here, we provide a detailed discussion of similar works to our method, accentuating limitations in existing methods and noting how our approach improves upon them.

## A.1. Upper Bounds: Comparison with AYS and DMN

First, we discuss our relationship with Align Your Steps (AYS) (Sabour et al., 2024) and DMN (Xue et al., 2024), two methods for learning optimized discretization schedules for DMs by minimizing upper bounds of various forms of error; however, minimizing these upper bounds provides no guarantee of actually minimizing the true global error. Additionally, because these methods only focus on selecting discretization schedules, they fail to fully explore the full design space of the DM sampler.

**DMN**  In DMN, Xue et al. (2024) minimizes an upper bound for the global error by optimizing only over the discretization schedules without considering the influence of the ODE solver method or the neural network; this bound is constructed solely by the chosen schedules for $\sigma_t$ and $\alpha_t$ that govern the SNR. Moreover, it makes a strong assumption that the prediction error of the score network is uniformly bounded by a small constant, which often fails to be the case (Zhang & Chen, 2023).

**AYS**  In AYS, Sabour et al. (2024) constructs an upper bound on the KL divergence between the true diffusion SDE solution distribution and the observed sampling distribution. They minimize this bound through an expensive Monte Carlo procedure and require bespoke numerical solutions, such as early stopping and a large batch size, to ensure stable optimization. More generally, both methods optimize an upper bound to their specific notions of error, which fails to guarantee minimization of the actual global error.

## A.2. Local Truncation Error: Comparison with DPM-Solver-v3, GITS, AMED-Plugin, ΠA, and Bespoke Solvers

Here, we provide discussion of a variety of works, which learn discretization schedules (Chen et al., 2024), solver coefficients (Zheng et al., 2023; Zhang et al., 2023), or a combination of both (Zhou et al., 2024a; Shaul et al., 2024a) by minimizing various forms of local truncation error. As previously discussed, we emphasize that such an optimization pattern is insufficient in ensuring that the global error is minimized, as well as method-specific differences or pathologies.

**DPM-Solver-v3**  DPM-Solver-v3 (Zheng et al., 2023) is descended from a remarkable family of exponential integrator-based work (Lu et al., 2022a; Zheng et al., 2023). Notably, DPM-Solver-v3 computes *empirical model statistics*, or EMS, that define coefficients that minimize the first-order discretization error produced from a Taylor expansion of their solver formulation. Interestingly, while these methods only minimize the first-order error, they are also used in higher-order versions of DPM-Solver-v3. Crucially, however, the EMS are calculated to ensure local truncation error control and ultimately provide global error control of the form $\mathcal{O}(h^k)$ given an $k$-th order predictor and maximum step size $h$. As a result, DPM-Sovler-v3 suffers from the same pathologies as other traditional solvers that aim to control the local truncation error when the step size becomes large. Additionally, Zheng et al. (2023) only learns the solver coefficients, leaving half of the sampler design space on the table.

**GITS**  Similarly, GITS (Chen et al., 2024), a method that uses DP-based search to select and optimized sequence of discretization steps for a DM, seeks to minimize the local truncation error of a student sampler. However, as discussed in Section 2.2, minimizing the local truncation error provides no guarantees for a bound on the global error, particularly in the low NFE regime; their algorithm reflects as much, as it assumes scaling of the local truncation error in order to obtain an estimate of the global error. Additionally, their method of selecting the discretization steps is agnostic to the specific choice of ODE solver used by the student sampler.

**AMED-Plugin**  AMED-Plugin (Zhou et al., 2024a) is a recently proposed approach that learns both coefficients and time step for existing solvers by selecting intermediate time steps within an existing discretization schedule and applying a learned scaling factor when using the intermediate point in an ODE solver; they do so by learning an additional "designer" neural network on top of the bottleneck feature extracted from a UNet-based score network. A reasonable interpretation of AMED-Plugin is that it learns half of the time steps used in a sampling procedure that can be used on top of many common solvers; accordingly, it does not take full advantage of the sampler design space, e.g. selecting all solver coefficients and time steps. Moreover, the neural network used in AMED-Plugin is also trained to minimize truncation error by matching

teacher trajectories along intermediate points, resulting in the same limitations as in Section 2.2. It also requires longer training time, which is likely attributable to the more expressive number of parameters being learned.

**ΠA**   ΠA (Zhang et al., 2023) is an approach that learns specific solver coefficients of different traditional solvers by minimizing the MSE between a student trajectory, requiring relatively minimal optimization costs. Similar to earlier critiques, matching the teacher trajectory can still learn pathologies along the teacher trajectory that are corrected with the benefit of additional NFEs but are ill-suited for the sutdent solver. Moreover, this approach only learns coefficients, failing to exploit the full design space; as a result, their quantitative performance is not as good as S4S.

**Bespoke Solvers**   Bespoke solver (Shaul et al., 2024a) is a solver distillation method that effectively learns both time steps and coefficients by constructing and minimizing an upper bound for the global error; in practice, this bound essentially just results in minimizing the sum of the local truncation error from a teacher solver. As a result, though it makes use of the full sampler design space, it also seeks to minimize a sub-optimal objective.

### A.3. Minimizing Global Error: Comparison with BNS and LD3

Finally, we discuss two approaches that seek to directly minimize the global error, either by learning discretization steps (Tong et al., 2024) or by learning both time steps and solver coefficients (Shaul et al., 2024b). While both of these objectives are aligned with our approach, they fail to achieve optimal performance in particular ways.

**BNS**   Bespoke Non-stationary Solvers (BNS) (Shaul et al., 2024b) directly minimizes the global error, in this case PSNR, based solely on the outputs of the student and teacher DM sampler. While this is aligned with our approach, they have three key limitations. First, their solvers, which are essentially learned versions of linear multi-step methods, have *maximal* order; that is, they allow the earliest predictions of the diffusion model to serve as gradient information even at very late time steps. Essentially, these solvers are $N$-step methods that leverage information from the full trajectory. Past work (Zheng et al., 2023) and our own ablations demonstrate that attempting to use methods with too much influence from past steps can result in instability in the ODE trajectories. Second, in the low NFE regime, BNS still has a relatively small number of parameters, which makes their objective difficult to optimize and results in solvers that likely are underfitted; we rectify such issues with our relaxed objective. Third, BNS optimizes all parameters simultaneously, which results in a complex optimization landscape irrespective of the whether the student model is adequately parametrized. In contrast, our approach uses alternating minimization to improve the stability of our overall optimization and iteratively solve optimization problems with easier loss landscape.

**LD3**   LD3 (Tong et al., 2024) uses a gradient-based method for learning a discretization schedule that minimizes the global error. Moreover, they also make use of a relaxed objective that makes their optimization problem easier when using a relatively small number of parameters. However, LD3 similarly fails to make use of the second half or the DM sampler design space, which yields a significant improvement in performance.

## B. Local Error Control in ODE Solvers

For completeness, we provide some details truncation error control for traditional ODE solver methods; significantly more details can be found in Lu et al. (2022a).

### B.1. Taylor Series Derivation

Here, we provide brief details of the derivation of the Taylor series and its low-order derivative terms, as referenced in Section 2.2. For further details and the most informative description of the relationship of diffusion ODE solvers to the low-order Taylor approximation, see Lu et al. (2022a;b); our explanation is essentially derived from their analysis. Recall that an exact solution for the diffusion ODE in its $\lambda$ parametrization can be given by

$$\mathbf{x}_{t_i} = \frac{\alpha_{t_i}}{\alpha_{t_{i-1}}}\mathbf{x}_{t_{i-1}} - \alpha_{t_i}\int_{\lambda_{t_{i-1}}}^{\lambda_{t_i}} e^{-\lambda}\hat{\boldsymbol{\epsilon}}_{\boldsymbol{\theta}}(\hat{\mathbf{x}}_{\lambda}, \lambda)d\lambda, \tag{10}$$

where $\hat{\mathbf{x}}_\lambda$ and $\hat{\boldsymbol{\epsilon}}_{\boldsymbol{\theta}}(\hat{\mathbf{x}}_\lambda, \lambda)$ denote the reparametrized forms of $\mathbf{x}_t$ and $\boldsymbol{\epsilon}_{\boldsymbol{\theta}}(\mathbf{x}_t, t)$ in the $\lambda$ domain. To compute $\mathbf{x}_{t_i}$, we must approximate the integral in Eq. (10); to do so, consider a Taylor expansion of $\hat{\boldsymbol{\epsilon}}_{\boldsymbol{\theta}}(\hat{\mathbf{x}}_\lambda, \lambda)$ as

$$\hat{\boldsymbol{\epsilon}}_{\boldsymbol{\theta}}(\hat{\mathbf{x}}_\lambda, \lambda) = \sum_{n=0}^{k-1} \frac{(\lambda - \lambda_{t_{i-1}})^n}{n!} \hat{\boldsymbol{\epsilon}}_{\boldsymbol{\theta}}^{(n)}(\hat{\mathbf{x}}_{\lambda_{t_{i-1}}}, \lambda_{t_{i-1}}) + \mathcal{O}((\lambda - \lambda_{t_{i-1}})^k)$$

Additionally, define the functions

$$\varphi_k(z) := \int_0^1 e^{(1-\delta)z} \frac{\delta^{k-1}}{(k-1)!} d\delta, \quad \varphi_0(z) = e^z,$$

which are common terms in exponential integrator methods (Hochbruck & Ostermann, 2010). Note that we have that $\varphi_k(0) = 1/k!$ with recurrence relation $\varphi_{k+1}(k) = (\varphi_k(z) - \varphi_k(0))/z$. Substituting the Taylor expansion into Eq. (10) and defining $h := \lambda_{t_i} - \lambda_{t_{i-1}}$ gives:

$$
\begin{aligned}
\mathbf{x}_{t_i} &= \frac{\alpha_{t_i}}{\alpha_{t_{i-1}}} \mathbf{x}_{t_{i-1}} - \alpha_{t_i} \int_{\lambda_{t_{i-1}}}^{\lambda_{t_i}} e^{-\lambda} \hat{\boldsymbol{\epsilon}}_{\boldsymbol{\theta}}(\hat{\mathbf{x}}_\lambda, \lambda) d\lambda \\
&= \frac{\alpha_{t_i}}{\alpha_{t_{i-1}}} \mathbf{x}_{t_{i-1}} - \alpha_{t_i} \int_{\lambda_{t_{i-1}}}^{\lambda_{t_i}} e^{-\lambda} \left( \sum_{n=0}^{k-1} \frac{(\lambda - \lambda_{t_{i-1}})^n}{n!} \hat{\boldsymbol{\epsilon}}_{\boldsymbol{\theta}}^{(n)}(\hat{\mathbf{x}}_{\lambda_{t_{i-1}}}, \lambda_{t_{i-1}}) + \mathcal{O}\left(h^k\right) \right) d\lambda \\
&= \frac{\alpha_{t_i}}{\alpha_{t_{i-1}}} \mathbf{x}_{t_{i-1}} - \alpha_{t_i} \left( \frac{\sigma_{t_i}}{\alpha_{t_i}} \sum_{n=0}^{k-1} h^{n+1} \varphi_{n+1}(h) \hat{\boldsymbol{\epsilon}}_{\boldsymbol{\theta}}^{(n)}(\hat{\mathbf{x}}_{\lambda_{t_{i-1}}}, \lambda_{t_{i-1}}) + \mathcal{O}\left(h^{k+1}\right) \right) \\
&= \frac{\alpha_{t_i}}{\alpha_{t_{i-1}}} \mathbf{x}_{t_{i-1}} - \sigma_{t_i} \sum_{n=0}^{k-1} h^{n+1} \varphi_{n+1}(h) \hat{\boldsymbol{\epsilon}}_{\boldsymbol{\theta}}^{(n)}(\hat{\mathbf{x}}_{\lambda_{t_{i-1}}}, \lambda_{t_{i-1}}) + \mathcal{O}\left(h^{k+1}\right)
\end{aligned}
$$

Taking $\psi_n(h) = h^{n+1} \varphi_{n+1}(h)$ yields the expression in Eq. (4). Moreover, note that

$$\varphi_1(h) = \frac{e^h - 1}{h}, \quad \varphi_2(h) = \frac{e^h - h - 1}{h^2}, \quad \varphi_3(h) = \frac{e^h - h^2/2 - 1}{h^3},$$

and accordingly we factor out an $e^h - 1$ to receive

$$\mathbf{x}_{t_i} = \frac{\alpha_{t_i}}{\alpha_{t_{i-1}}} \mathbf{x}_{t_{i-1}} - \sigma_{t_i}(e^h - 1) \sum_{n=0}^{k-1} c_n(h) \hat{\boldsymbol{\epsilon}}_{\boldsymbol{\theta}}^{(n)}(\hat{\mathbf{x}}_{\lambda_{t_{i-1}}}, \lambda_{t_{i-1}}) + \mathcal{O}\left(h^{k+1}\right).$$

where $c(h)$ captures the appropriate coefficient of each $\hat{\boldsymbol{\epsilon}}_{\boldsymbol{\theta}}^{(n)}$. This essentially captures the desired formulation we provide: a given ODE solver method approximates the $\hat{\boldsymbol{\epsilon}}_{\boldsymbol{\theta}}^{(n)}$ terms, we capture this approximation using $\Delta_i$ and ignore the higher-order Taylor terms.

### B.2. Regularity Conditions for Local Truncation Error Control

In general, three regularity conditions (Lu et al., 2022a;b; Zheng et al., 2023) are required for ensuring that the local truncation error can be bounded in common diffusion ODE solvers:

1. The derivatives $\hat{\boldsymbol{\epsilon}}_{\boldsymbol{\theta}}^{(n)}$ in Eq. (4) exist and are continuous for all $0 \leq n \leq k$.

2. The score network $\boldsymbol{\epsilon}_{\boldsymbol{\theta}}$ is Lipschitz in its first parameter $\mathbf{x}$.

3. The maximum step size $h_{max}$ is $\mathcal{O}(1/N)$, where $N$ is the number of discretization steps.

These assumptions break down in the following ways:

1. The derivatives of the noise prediction model $\hat{\boldsymbol{\epsilon}}_{\boldsymbol{\theta}}^{(n)}$ cannot be guaranteed to exist or be continuous, since neural networks trained with standard optimizers like SGD or Adam do not enforce smoothness constraints on the learned function. While techniques like spectral normalization (Miyato et al., 2018) can help control Lipschitz constants, they do not ensure differentiability.

| Solver Type | $\Delta_i(\phi)$ | $\phi$ | NFEs per Step | # Params. |
|---|---|---|---|---|
| LMS | $\displaystyle\sum_{j=1}^{k} b_{j,i}\mathbf{x}_{\boldsymbol{\theta}}(\tilde{\mathbf{x}}_{t_{i-j}}, t_{i-j})$ | $\{b_{j,i}\}$ | 1 | $k(2N+1-k)/2$ |
| SS | $\displaystyle\sum_{j=1}^{k} b_{j,i}\kappa_j, \; \kappa_j = \mathbf{x}_{\boldsymbol{\theta}}\left(\tilde{\mathbf{x}}_{t_{i-1}} + \sum_{l=1}^{j-1} a_{j,i,l}\kappa_l, t_{i-1} + c_{j,i}\right)$ | $\{b_{j,i}, a_{j,i,l}, c_{j,i}\}$ | $k$ | $(k^2+k-1)N$ |
| LMS + PC | $\displaystyle\sum_{j=1}^{k} a_{j,i}^c \mathbf{x}_{\boldsymbol{\theta}}(\tilde{\mathbf{x}}_{t_{i-j}}, t_{i-j})$ | $\{b_{j,i}\} + \{a_{j,i}^c\}$ | 1 | $k(2N+1-k)$ |

*Table 7.* We apply S4S to three types of diffusion ODE solvers; we show their increment ($\Delta_i$), learnable parameters, number of NFEs per step, and total parameter count over $N+1$ steps. By default, we use a linear multi-step predictor for the PC method, so $\{a_{j,i}^c\}$ refer to coefficients during the correction step, and the total set of learnable parameters accounts for the underlying multi-step predictor.

2. The Lipschitz condition on $\epsilon_{\boldsymbol{\theta}}$ is typically violated in practice, as modern score networks use architectures like U-Nets that can have very large Lipschitz constants. Even with normalization techniques, these constants often scale poorly with network depth and width.

3. The step size restriction $h_{max} = \mathcal{O}(1/N)$ forces a trade-off between computational cost and numerical accuracy that may be unnecessarily conservative in many regions of the trajectory where the ODE is well-behaved.

These theoretical limitations help explain why practical implementations often deviate from the idealized analysis. In particular, alternative methods for local truncation error control (Zhang & Chen, 2023; Chen et al., 2024) can achieve good empirical performance despite violating these assumptions, suggesting that weaker conditions may be sufficient in practice.

### B.3. Local Error Control

A number of related works (Chen et al., 2024; Zhang et al., 2023; Shaul et al., 2024a) recommend *matching* the trajectory of the teacher solver. In our setting, given an intermediate point $\tilde{\mathbf{x}}_i^*$ from the teacher solver, this would require optimizing an objective of the form:

$$\min_{\phi} \|d(\tilde{\mathbf{x}}_i^{\phi}, \tilde{\mathbf{x}}_i^*)\|$$

for all $i$ in $[N]$, either simultaneously or iteratively for each $i$. Nonetheless, across many teacher trajectories, many solvers have pathological behavior that is corrected in regimes with large numbers of NFEs. For example, Figure 9 in Zhou et al. (2024b) demonstrates such an example: as the guidance scale increases, the teacher trajectories become increasingly pathological, but benefit from correcting errors made in early steps. However, by training a student solver with few NFEs to match such a trajectory on overlapping points with the teacher solver, it can learn these same pathologies that are resolved in the teacher by a larger number of NFEs.

## C. Generalized Formulation of Diffusion ODE Solvers

### C.1. Data Prediction Solver Instantiation

While we focus in the main paper on generalized versions of ODE solvers in terms of noise prediction, we also provide a general expression in terms of the data prediction model. Note that the general form of the exact solution to the diffusion ODE under parametrization by the data prediction model is

$$\mathbf{x}_t = \frac{\sigma_t}{\sigma_s}\mathbf{x}_s + \sigma_t \int_{\lambda_s}^{\lambda_t} e^{\lambda}\hat{\mathbf{x}}_{\theta}(\hat{\mathbf{x}}_{\lambda}, \lambda)d\lambda$$

Therefore, we just need to take a Taylor approximation of the integral, as we did in Appendix B.1. This results in a general expression for a diffusion ODE as

$$\tilde{\mathbf{x}}_{t_i} = \frac{\sigma_{t_i}}{\sigma_{t_{i-1}}}\tilde{\mathbf{x}}_{t_{i-1}} - \alpha_{t_i}(e^{-h_i} - 1)\Delta_i^{\mathbf{x}}(\phi)$$

We display the equivalent definitions for $\Delta_i^{\mathbf{x}}(\phi)$ in Table 7.

## C.2. Constant Coefficients in Diffusion ODE Solvers

Coefficients in diffusion model solvers are not "inherently" constant; whether they are constant or not depends on the choice of discretization schedule and design decisions in the solver. For example, the iPNDM solver (Zhang & Chen, 2023) demonstrates this principle clearly - after its initial warmup period, it settles into using constant coefficients for subsequent steps. This design choice provides computational efficiency while maintaining numerical stability. The solver achieves this by carefully transitioning from variable coefficients during the warmup phase to fixed values that work well across the remaining time steps.

Similarly, DPM-Solver++ (Lu et al., 2022b) multi-step methods can be viewed through the lens of constant coefficients, particularly in their higher-order variants. This perspective helps explain their computational efficiency, as the coefficients don't need to be recalculated at each step, while still maintaining high-order accuracy in solving the diffusion ODE.

# D. Relaxed Objective

## D.1. Theoretical Guarantee

Here, we briefly restate the theoretical guarantee for the relaxed objective presented in Eq. (7); this guarantee was provided by Tong et al. (2024).

**Theorem D.1.** *Let $\Psi_*$ and $\Psi_\phi$ be a teacher and student ODE solver each with noise distribution $\mathcal{N}(0, \sigma_1^2 \mathbf{I}) \in \mathbb{R}^d$, and with, respectively, distributions $q$ and $p_\phi$. Assume both $\Psi_*$ and $\Psi_\phi$ are invertible. Let $r > 0$, if the objective from Equation (6) has an optimal solution $\phi^*$ for $r$ with objective value 0, we have*

$$D_{\mathrm{KL}}(q(\mathbf{x}) \parallel p_{\phi^*}(\mathbf{x})) \leq \frac{r^2}{2} + r\sqrt{d+1} + \mathbb{E}_{\mathbf{x} \sim q(\mathbf{x})} \| C(\Psi_*(\mathbf{x})) - C(\Psi_{\phi^*}(\mathbf{x})) \|, \tag{11}$$

*where $C(\Psi_{\phi^*}(\mathbf{x})) = \log | \det J_{\Psi_{\phi^*}} (\Psi_{\phi^*}^{-1}(\mathbf{x})) |$.*

Below, we provide a provide a brief overview of the proof; see Tong et al. (2024)[A.1] for further details.

*Proof.* We proceed in several steps:

**Step 1: Initial Setup.** By assuming the invertibility of the solvers and the loss of Equation (6) having an optimal (zero loss and satisfying all $r\sigma_T$-ball constraints) solution $\phi^*$, we have for every $\mathbf{x} \sim q(\mathbf{x})$ exactly one $\mathbf{b}$ with $\Psi_*^{-1}(\mathbf{x}) = \mathbf{b}$ and exactly one corresponding $\mathbf{a}$ with $\Psi_{\phi^*}^{-1}(\mathbf{x}) = \mathbf{a}$. Moreover, since $\mathbf{a}$ is an optimal and therefore feasible solution, we have $\mathbf{a} \in B(\mathbf{b}, r\sigma_T)$ and thus $\|\mathbf{a} - \mathbf{b}\|_2 \leq r\sigma_T$.

**Step 2: KL Divergence Expansion.** Using the density function of the normal distribution, we can write:

$$\mathbb{E}_{\mathbf{x} \sim q(\mathbf{x})} \left[ \log \left( \frac{q(\mathbf{x})}{p_\phi(\mathbf{x})} \right) \right] = \mathbb{E}_{\mathbf{x} \sim q(\mathbf{x})} \left[ \log \left( \frac{\mathcal{N}(\mathbf{b}) \left| \det \frac{d\Psi_*(\mathbf{b})}{d\mathbf{b}} \right|^{-1}}{\mathcal{N}(\mathbf{a}) \left| \det \frac{d\Psi_{\phi^*}(\mathbf{a})}{d\mathbf{a}} \right|^{-1}} \right) \right]$$

$$= \mathbb{E}_{\mathbf{x} \sim q(\mathbf{x})} \left[ \log(\mathcal{N}(\mathbf{b})) + \log \left( \left| \det \frac{d\Psi_*(\mathbf{b})}{d\mathbf{b}} \right|^{-1} \right) - \log(\mathcal{N}(\mathbf{a})) - \log \left( \left| \det \frac{d\Psi_{\phi^*}(\mathbf{a})}{d\mathbf{a}} \right|^{-1} \right) \right]$$

**Step 3: Normal Distribution Terms.** The normal distribution terms can be written explicitly:

$$\mathbb{E}_{\mathbf{x} \sim q(\mathbf{x})} \left[ \log \left( \frac{\prod_{i=1}^d \frac{1}{\sigma_T \sqrt{2\pi}} \exp \left( -\frac{1}{2} \frac{b_i^2}{\sigma_T^2} \right)}{\prod_{i=1}^d \frac{1}{\sigma_T \sqrt{2\pi}} \exp \left( -\frac{1}{2} \frac{a_i^2}{\sigma_T^2} \right)} \right) \right]$$

**Step 4: Clever Substitution.** We rewrite $a_i = b_i + \epsilon_i$ for $\epsilon_i \in \mathbb{R}$. This gives:

$$\mathbb{E}_{\mathbf{x} \sim q(\mathbf{x})} \left[ \frac{1}{2\sigma_T^2} \sum_{i=1}^d (2\epsilon_i b_i + \epsilon_i^2) \right] = \frac{1}{\sigma_T^2} \mathbb{E}_{\mathbf{x} \sim q(\mathbf{x})} \left[ \sum_{i=1}^d \epsilon_i b_i \right] + \frac{1}{2\sigma_T^2} \mathbb{E}_{\mathbf{x} \sim q(\mathbf{x})} \left[ \sum_{i=1}^d \epsilon_i^2 \right]$$

**Step 5: Final Bound on First Term.** Since $\|\mathbf{a} - \mathbf{b}\|_2 \le r\sigma_T$, we have that $\sum_{i=1}^{d}(a_i - b_i)^2 \le r^2\sigma_T^2$ and with $a_i = b_i + \epsilon_i$, we have $\sum_{i=1}^{d} \epsilon_i^2 \le r^2\sigma_T^2$.

Therefore:

$$\frac{1}{2\sigma_T^2}\mathbb{E}_{\mathbf{x}\sim q(\mathbf{x})}\left[\sum_{i=1}^{d} \epsilon_i^2\right] \le \frac{1}{2\sigma_T^2}\mathbb{E}_{\mathbf{x}\sim q(\mathbf{x})}[r^2\sigma_T^2] = \frac{r^2}{2}$$

**Step 6: Independence and Cauchy-Schwarz.** The last equality follows from the independence of random variables in the multivariate distribution. Applying the Cauchy-Schwarz inequality:

$$\frac{1}{\sigma_T^2}\mathbb{E}_{\mathbf{x}\sim q(\mathbf{x})}\left[\sum_{i=1}^{d} \epsilon_i b_i\right] \le \frac{1}{\sigma_T^2}\mathbb{E}_{\mathbf{x}\sim q(\mathbf{x})}\left[\left(\sum_{i=1}^{d} \epsilon_i^2\right)^{1/2}\left(\sum_{i=1}^{d} b_i^2\right)^{1/2}\right]$$

$$\le \frac{1}{\sigma_T^2}\mathbb{E}_{\mathbf{x}\sim q(\mathbf{x})}\left[r\sigma_T\left(\sum_{i=1}^{d} b_i^2\right)^{1/2}\right]$$

$$= \frac{r}{\sigma_T}\mathbb{E}_{\mathbf{b}\sim\mathcal{N}(0,\sigma_T^2\mathbf{I})}\left[\left(\sum_{i=1}^{d} b_i^2\right)^{1/2}\right]$$

**Step 7: Chi-squared Distribution.** Since $b_i \overset{\text{i.i.d.}}{\sim} \mathcal{N}(0, \sigma_T^2)$, the sum of squares follows a Chi-squared distribution scaled by $\sigma_T^2$:

$$\sum_{i=1}^{d} b_i^2 \sim \sigma_T^2\chi_d^2$$

This allows us to write:

$$\frac{r}{\sigma_T}\mathbb{E}\left[\left(\sum_{i=1}^{d} b_i^2\right)^{1/2}\right] = \frac{r}{\sigma_T}\mathbb{E}\left[\sigma_T\sqrt{\chi_d^2}\right] = r\sigma_T\mathbb{E}\left[\sqrt{\chi_d^2}\right] = r\sqrt{2}\frac{\Gamma\left(\frac{d+1}{2}\right)}{\Gamma\left(\frac{d}{2}\right)}$$

**Step 8: Gautschi's Inequality.** Applying Gautschi's inequality:

$$\frac{\Gamma\left(\frac{d+1}{2}\right)}{\Gamma\left(\frac{d}{2}\right)} \le \sqrt{\frac{d+1}{2}}$$

This gives us:

$$r\sqrt{2}\frac{\Gamma\left(\frac{d+1}{2}\right)}{\Gamma\left(\frac{d}{2}\right)} \le r\sqrt{2}\sqrt{\frac{d+1}{2}} = r\sqrt{d+1}$$

**Step 9: Final Bound.** Combining all terms from Equation (8), we obtain our final bound:

$$D_{\text{KL}}(q(\mathbf{x}) \parallel p_{\phi^*}(\mathbf{x})) \le \frac{r^2}{2} + r\sqrt{d+1} + \mathbb{E}_{\mathbf{x}\sim q(\mathbf{x})}[\|C(\Psi_*(\mathbf{x})) - C(\Psi_{\phi^*}(\mathbf{x}))\|]$$

where $C(\Psi_{\phi^*}(\mathbf{x})) = \log|\det J_{\Psi_{\phi^*}}(\Psi_{\phi^*}^{-1}(\mathbf{x}))|$. $\qquad\square$

Evaluating whether the solver is invertible is difficult to characterize in practice. We note, however, that LMS solvers can at least be represented in matrix form, as they scale a linear combination of previous evaluations of the model. Accordingly, if only the coefficients are learned, then the LMS solver can be made invertible by the transform $A \mapsto A + \epsilon\mathbf{I}$ for a sufficiently small, non-zero $|\epsilon|$.

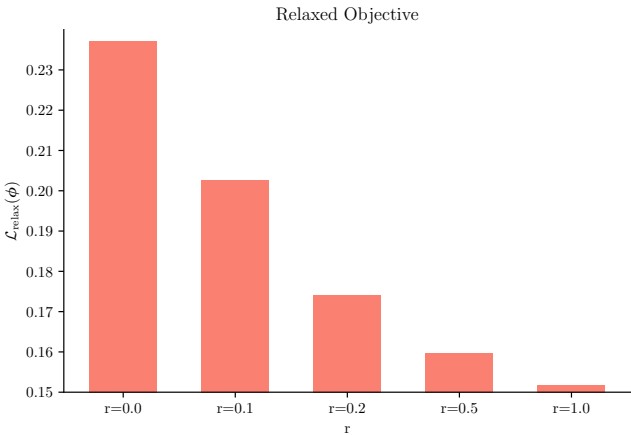

*Figure 3.* Values of $\mathcal{L}_{\text{relax}}$ as we expand $r$. As $r$ increases, the objective becomes easier to optimize, thereby validating the utility of the relaxed objective in making an easier optimization problem for learning solver coefficients.

### D.2. Easier Objective

We also hope to verify that the relaxed objective is indeed easier to optimize. We characterize this by running an experiment on CIFAR-10: we optimize the S4S coefficients initialized at iPNDM with logSNR discretization and characterize the empirical loss of Eq. (7) as $r$ increases. We affirmatively verify this in Figure 3.

## E. Parametrization of Solver Discretization Steps

We parameterize the two versions of our time steps, $t_i^{\boldsymbol{\xi}}$ and $t_i^c$, in two distinct stages described below.

### E.1. General Time Steps

Given a learnable vector $\boldsymbol{\xi} \in \mathbb{R}^{N+1}$, we construct each time step $t_i^{\boldsymbol{\xi}}$ through a two-stage process. First, we apply a cumulative softmax operation to ensure strict monotonicity:

$$\tau'_{\boldsymbol{\xi}}(i) = \sum_{n=i}^{N} \text{softmax}(\boldsymbol{\xi})[n]$$

We then apply a linear rescaling to map these values to the interval $[t_{\min}, T]$:

$$t_i^{\boldsymbol{\xi}} = \frac{\tau'_{\boldsymbol{\xi}}(i) - \tau'_{\boldsymbol{\xi},\min}}{\tau'_{\boldsymbol{\xi},\max} - \tau'_{\boldsymbol{\xi},\min}}(T - t_{\min}) + t_{\min}$$

This construction ensures that $t_0^{\boldsymbol{\xi}} = T > t_1^{\boldsymbol{\xi}} > \cdots > t_N^{\boldsymbol{\xi}} = t_{\min}$, and ultimately provides the foundation for determining step sizes and signal-to-noise ratio parameters, as described in the main text.

### E.2. Decoupled Time Steps

Following the parameterization of $\{t_i^{\boldsymbol{\xi}}\}_{i=0}^{N+1}$, we now construct the decoupled time steps $\{t_i^c\}_{i=0}^{N}$ that are used as input to the score network. Specifically, we define each decoupled time step $t_i^c$ as

$$t_i^c = t_i^{\boldsymbol{\xi}} + \boldsymbol{\xi}_i^c$$

where $\boldsymbol{\xi}^c \in \mathbb{R}^{N+1}$ is a learnable offset vector. For numerical stability, we constrain the magnitude of the decoupled offsets $\boldsymbol{\xi}^c$. Let $\Delta t_i = |t_{i+1}^{\boldsymbol{\xi}} - t_i^{\boldsymbol{\xi}}|$ be the gap between consecutive time steps. We define the maximum allowed offset as $\delta = \alpha \min_i \Delta t_i$, where $\alpha > 0$ is a hyperparameter. The final decoupled time steps are then given by:

$$t_i^c = \begin{cases} t_i^{\boldsymbol{\xi}} & \text{if } i \in \{0, N\} \\ t_i^{\boldsymbol{\xi}} + \text{clip}(\boldsymbol{\xi}_i^c, [-\delta, \delta]) & \text{otherwise} \end{cases}$$

where $\text{clip}(x, [a, b])$ clamps the value of $x$ to the interval $[a, b]$. This ensures that the endpoints remain fixed while intermediate steps can only shift by a fraction of the smallest step size.

# F. Additional Implementation Details

## F.1. Pseudocode for S4S-Alt

Here, we describe the pseudocode for S4S-Alt, which strongly resembles that of S4S. However, we emphasize that we use the same value of $r$ that bounds the allowed deviation of the initial noise condition in *both* optimization objectives. We do this because both objectives must *share* the same allowable distribution of the noise; otherwise, starting from different initial conditions in different parts of the overall optimization makes learning the effective parameters much more difficult. Additionally, using S4S-Alt generally requires significantly more examples relative to S4S, as we hope to ensure that both sets of parameters do not begin to overfit.

---

**Algorithm 2** S4S-Alt

---

**Require:** Coefficient parameters $\phi$, discretization step parameters $\Xi$, student solver $\Psi_{\phi,\Xi}$, teacher solver $\Psi^*$, distance metric $d$, number of alternating steps $K$, and $r$.
1: $\mathcal{D} \leftarrow \{(\mathbf{x}'_T, \mathbf{x}_T, \Psi^*(\mathbf{x}_T)) \mid \mathbf{x}_T \sim \mathcal{N}(\mathbf{0}, \tilde{\sigma}^2 \mathbf{I}), \mathbf{x}'_T = \mathbf{x}_T\}$ *// Generate data $\mathcal{D}_k$*
2: $k \leftarrow 1$
3: **for** $k = 1, \ldots, K$ **do**
4:     **while** not converged **do**
5:         $(\mathbf{x}'_T, \mathbf{x}_T, \Psi^*(\mathbf{x}_T)) \sim \mathcal{D}$
6:         $\mathcal{L}(\phi, \mathbf{x}'_T) = d(\Psi_{\phi,\Xi}(\mathbf{x}'_T), \Psi^*(\mathbf{x}_T))$ subject to $\mathbf{x}'_T \in B(\mathbf{x}_T, r\sigma_T)$
7:         Update $\phi$ and $\mathbf{x}'_T$ using the corresponding gradients $\nabla \mathcal{L}(\phi, \mathbf{x}'_T)$
8:         $\mathbf{x}'_T \leftarrow \mathbf{x}_T + \mathbf{1}[\|\mathbf{x}'_T - \mathbf{x}_T\|_2 > r] \cdot r \frac{\mathbf{x}'_T - \mathbf{x}_T}{\|\mathbf{x}'_T - \mathbf{x}_T\|_2}$ *// Projected SGD*
9:         Update $\mathcal{D}$ with the new $\mathbf{x}'_T$
10:     **end while**
11:     **while** not converged **do**
12:         $(\mathbf{x}'_T, \mathbf{x}_T, \Psi^*(\mathbf{x}_T)) \sim \mathcal{D}$
13:         $\mathcal{L}(\Xi, \mathbf{x}'_T) = d(\Psi_{\phi,\Xi}(\mathbf{x}'_T), \Psi^*(\mathbf{x}_T))$ subject to $\mathbf{x}'_T \in B(\mathbf{x}_T, r\sigma_T)$
14:         Update $\Xi$ and $\mathbf{x}'_T$ using the corresponding gradients $\nabla \mathcal{L}(\Xi, \mathbf{x}'_T)$
15:         $\mathbf{x}'_T \leftarrow \mathbf{x}_T + \mathbf{1}[\|\mathbf{x}'_T - \mathbf{x}_T\|_2 > r] \cdot r \frac{\mathbf{x}'_T - \mathbf{x}_T}{\|\mathbf{x}'_T - \mathbf{x}_T\|_2}$ *// Projected SGD*
16:         Update $\mathcal{D}$ with the new $\mathbf{x}'_T$
17:     **end while**
18: **end for**

---

## F.2. Efficient Computational Techniques

To optimize memory usage during training, we employ gradient rematerialization when computing $\nabla_\phi \Psi_\phi(\mathbf{x}'_T)$. Rather than storing all intermediate neural network activations, which would incur $\mathcal{O}(N)$ memory overhead with respect to the number of parameters, we recompute them on the fly during backpropagation. This approach follows Tong et al. (2024) and Watson et al. (2021), trading increased computation time for reduced memory requirements. Specifically, we rematerialize calls to the pretrained score network $\epsilon_\theta$ while maintaining the chain of denoised states in memory, allowing our method to scale to large diffusion architectures while maintaining reasonable batch sizes.

# G. Experiment Details

## G.1. Discretization Heuristics and Methods

We use four time discretization heuristics and three methods for adaptively selecting the discretization steps. Here, we consider time interval from $T$ to $\epsilon$ over which the ODE is solved with $N + 1$ total time steps; here, solving the ODE to $\epsilon$ rather than 0 helps with numerical stability.

### G.1.1. DISCRETIZATION HEURISTICS

**Time Uniform and Time Quadratic Discretization**   In the Time Uniform discretization schedule, we split the interval $[T, \epsilon]$ uniformly; this gives discretization schedule:

$$t_n = T + \frac{n}{N}(\epsilon - T)$$

for $n \in [N]$. Alternatively, the Time Quadratic schedule assigns each time step as

$$t_n = T + \left(\frac{n}{N}\right)^2 (\epsilon - T).$$

These schedules are popular for variance preserving-style DMs (Ho et al., 2020; Song et al., 2021a; Lu et al., 2022a).

**Time EDM Discretization**   Karras et al. (2022) propose a change of variables to $\kappa_t = \frac{\sigma_t}{\alpha_t}$ and creating a discretization schedule according to

$$t_n = t_\kappa \left( \left( \kappa_T^{1/\rho} + \frac{n}{N} \left( \kappa_\epsilon^{1/\rho} - \kappa_T^{1/\rho} \right) \right)^\rho \right)$$

where $t_\kappa$ is the inverse of $t \mapsto \kappa_t$, which exists as $\kappa_t$ is strictly monotone by the construction of $\sigma_t$, $\alpha_t$.

**Time log-SNR Discretization**   Alternatively, Lu et al. (2022a;b) propose a change of variables to $\lambda_t = \log(\alpha_t/\sigma_t)$ log-SNR domain and discretizing uniformly over the interval, i.e.

$$t_n = t_\lambda(\lambda_T + \frac{n}{N}(\lambda_\epsilon - \lambda_T))$$

where $t_\lambda$ is the inverse mapping of $t \mapsto \lambda_t$, which again exists because of strict monotonicity.

### G.1.2. DISCRETIZATION SCHEDULE SELECTION METHODS

**DMN**   DMN (Xue et al., 2024) constructs an optimization problem that creates an upper bound on the global error. Concretely, they model sequentially solving the diffusion ODE in terms of Lagrange approximations, construct an upper bound of the error on the assumption that the score network prediction error is uniformly upper bounded by a constant, and select a sequence of $\lambda_i$ that minimizes the derived upper bound.

**GITS**   GITS (Chen et al., 2024) is a method that uses DP-based search to select an optimized sequence of discretization steps for a DM that minimizes the deviation the diffusion ODE. They do so by calculating the local error incurred from estimating the next time step $t_i$ from the current step $t_{i-1}$ on a finely discretized search space of possible time steps. Once a cost matrix of all pair-wise costs is calculated, they then use a DP algorithm to select the lowest-cost sequence of steps given a number of NFEs. Intuitively, this approach seeks to take steps that are relatively large in regions of low curvature and smaller steps in regions with high curvature where the discretization error might be high.

**LD3**   LD3(Tong et al., 2024) seeks to learn a sequence of coefficients using the same parameterization as in Appendix E. They similarly try to minimize an objective over the global discretization error, often LPIPS.

### G.2. Practical Implementation

Here, we discuss important practical details that we use for both S4S and S4S-Alt. Most crucial is our choice of $r$ when optimizing our relaxed objective in both S4S and S4S-Alt. Let $m$ denote the total number of parameters learned in the student solver. Then in both S4S and S4S-Alt, we set $r \propto \frac{1}{m^{5/2}}$. This helps balance the solver's ability to learn the relaxed objective with the number of parameters that it has available.

In practice, for CIFAR-10, FFHQ, and AFHQv2, we use 700 samples for learning coefficients in S4S with a batch size of 20; when learning coefficients *and* time steps in S4S-Alt, we generally use 1400 samples as training data with a batch size of 40. We use 200 samples and 400 samples as a validation data set, respectively. For latent DMs, we use 600 samples for learning S4S with a batch size of 20 using gradient accumulation, and use a dataset of 1000 samples with batch size of 40 for S4S-Alt. We again use 200 samples and 400 samples as a validation data set, respectively. In both settings, we run S4S for 10 epochs, and S4S-Alt for $K$=8 alternating steps.

| Method | Order | NFE=3 | NFE=4 | NFE=6 | NFE=8 |
|---|---|---|---|---|---|
| DPM-Solver (2S) | 2 | - | 239.41 | 65.24 | 28.06 |
| DPM-Solver-S4S (2S) | 2 | - | 66.82 | 34.91 | 24.73 |
| DPM-Solver-S4S (3S) | 3 | 89.75 | - | 42.02 | - |
| iPNDM-S4S (3M) | 3 | **48.19** | **21.58** | **8.91** | **4.33** |

*Table 8.* FID of SS methods for S4S initialized at DPM-Solver. Although DPM-Solver-S4S achieves significant gains in FID, especially relative to its unlearned counterpart, it lags behind the simpler and much easier to optimize LMS methods.

For teacher solvers, in general we follow Tong et al. (2024) and select the best-performing solver at 20 NFE. This is UniPC with 20 NFE and logSNR discretization for CIFAR-10, FFHQ, and AFHQv2; UniPC with 20 NFE and time uniform discretization for LSUN Bedroom, UniPC with 10 NFE and time uniform discretization for Imagenet, and UniPC with GITS discretization at 10 steps for MS-COCO.

## H. Additional Results

### H.1. Single-Step Solvers

While in the main text we mainly focus on LMS methods, we also consider SS solver methods, in particular focusing on DPM-Solver (Lu et al., 2022a). In particular, we consider learnable equivalents of DPM-Solver (2S), a second-order method which uses a single intermediate step $u_1$ as well as $\tilde{\mathbf{x}}_{t_{i-1}}$ to estimate $\tilde{\mathbf{x}}_{t_i}$, and DPM-Solver (3S), which uses two intermediate steps $u_1$ and $u_2$ and is therefore a third-order method. Note that while the practical algorithmic approach for learning the SS coefficients is the same as that in the LMS setting, there are significantly more parameters that can be learned as compared to LMS or even PC methods. Consequently, the allowable radius $r$ of our relaxed objective is much smaller than its LMS counterparts.

Table 8 demonstrates our results on FFHQ using the logSNR discretization schedule. We compare against iPNDM-S4S as a baseline for LMS methods as well as to traditional DPM-Solver (2S). Here, we find that S4S similarly leads to significant gains for SS solvers, in fact even larger than the gains seen for LMS solvers. Nonetheless, despite the significant improvements attained by learning the solver coefficients, SS methods still lag behind their LMS counterparts. Intuitively, this is because SS methods have significantly more parameters to optimize. If $r$ is not chosen properly, then there is a significant chance that S4S *overfits* to the training dataset but fails to generalize well to the original noise distribution. Moreover, SS methods suffer from the fact that their effective step size is larger than that of LMS methods, i.e. for an equal number of NFEs, the step size of a $k$-step LMS method is $1/k$ the step size of the $k$-step SS method. As a result, for the core remaining parts of our experiments, we focus on LMS methods.

### H.2. Additional Ablations

We ablate several of the design decisions in our approach in Table 9. Specifically, we characterize the importance of time-dependent coefficients, the choice of LPIPS as our distance metric, and the use of the relaxed objective. We find that time-dependent coefficients significantly improves the performance of S4S and S4S-Alt; this is somewhat expected, since using a fixed set of coefficients for several iterations significantly decreases the number of learnable parameters. Additionally, we find that we still attain strong performance when using the $L_2$ loss in lieu of LPIPS. Finally, using our relaxed objective greatly improves performance, particularly in S4S with few NFEs, though with more NFEs the benefit decays as the optimization problem becomes less underparametrized.

#### H.2.1. S4S INITIALIZATION

A natural question to consider is the importance of the initialization heuristic used for S4S. Here, we consider the results of initializing an LMS method according to a standard Gaussian. We evaluate this initialization on CIFAR-10 and FFHQ with the logSNR discretization schedule; Table 10 contains our results for this evaluation. Although S4S initialized with standard Gaussian coefficients achieves meaningful improvements, it is nonetheless outperformed by initializing at existing solver methods.

|  | Setting | NFE=4 | NFE=5 |
|---|---|---|---|
| **S4S** | | | |
| Glob. $\phi$ | Full | 19.39 | 9.08 |
| | $r = 0$ | 19.62 | 10.17 |
| | $L_2$ | 33.25 | 15.13 |
| Time Dep. $\phi$ | Full | 33.34 | 19.39 |
| | $r = 0$ | 33.75 | 19.50 |
| | $L_2$ | 56.49 | 21.65 |
| **S4S-Alt** | | | |
| Glob. $\phi$ | Full | 27.99 | 13.32 |
| | $r = 0$ | 30.52 | 14.20 |
| | $L_2$ | 26.01 | 16.33 |
| Time Dep. $\phi$ | Full | 30.94 | 16.76 |
| | $r = 0$ | 31.04 | 17.64 |
| | $L_2$ | 38.73 | 21.97 |

*Table 9.* Ablations of model design decisions on CIFAR-10. To show the largest range of effects as a result of S4S, we use the EDM discretization schedule.

| Dataset | Method | NFE=3 | NFE=4 | NFE=5 | NFE=6 |
|---|---|---|---|---|---|
| CIFAR-10 | Gaussian-S4S (3M) | 91.84 | 42.17 | 25.61 | 11.93 |
| | iPNDM-S4S (3M) | 75.88 | 30.12 | 17.97 | 10.61 |
| | DPM-Solver-++-S4S (3M) | 93.58 | 40.18 | 22.21 | 11.04 |
| FFHQ | Gaussian-S4S (3M) | 81.44 | 44.91 | 24.83 | 15.01 |
| | iPNDM-S4S (3M) | 76.81 | 36.23 | 24.16 | 16.15 |
| | DPM-Solver-++-S4S (3M) | 86.39 | 45.89 | 22.52 | 13.78 |

*Table 10.* FID of LMS methods initialized with standard Gaussian coefficients and optimized using S4S compared against initialization at iPNDM or DPM-Solver++. We use the logSNR discretization heuristic for all samples. Gaussian-initialized S4S outperforms traditional ODE solvers, but nonetheless improves less than its solver-initialized counterparts.

### H.2.2. JOINT OPTIMIZATION OBJECTIVE AND DETAILS

Below, we describe the optimization objective and implementation details for learning the joint optimization objective, which learns both the solver coefficients and the time steps simultaneously. The pseudocode is essentially a restatement of that of S4S, but propagating the gradients to both sets of learnable coefficients. We use the same batch size

### H.2.3. TRAINING DATASET SIZE

We also ablate the significance of the training dataset size in S4S-Alt. We display these results for CIFAR-10 with 6 NFEs in Figure 4.

### H.2.4. ENSURING CONSISTENCY IN SINGLE-STEP SOLVERS

Although in general we abandon the notion of maintaining notions of local error control in our diffusion solvers, we consider an additional ablation for enforcing consistency in single-step solvers. That is, we ablate requiring the $b_{j,i}$ in single-step solvers sum to 1 for every $i$. We display these results in Table 11 – rather than consistency resulting in better global error, it in fact worsens our gobal error performance.

### H.3. Comparisons with Existing Learning-Based Methods

We continue our comparison of S4S and S4S-Alt against alternative learning-based methods. In Table 12, we characterize the computational requirements needed for each method for learning the solver, as well as its corresponding FID and number of NFEs needed to beat S4S-Alt.

**Algorithm 3** Joint Optimization Algorithm

---

**Require:** Coefficient parameters $\phi$, discretization step parameters $\boldsymbol{\xi}$, student solver $\Psi_{\phi,\Xi}$, teacher solver $\Psi^*$, distance metric $d$, and $r$.

1: $\mathcal{D} \leftarrow \{(\mathbf{x}'_T, \mathbf{x}_T, \Psi^*(\mathbf{x}_T)) \mid \mathbf{x}_T \sim \mathcal{N}(\mathbf{0}, \tilde{\sigma}^2 \mathbf{I}), \mathbf{x}'_T = \mathbf{x}_T\}$ *// Generate data $\mathcal{D}$*
2: **while** not converged **do**
3: $\quad (\mathbf{x}'_T, \mathbf{x}_T, \Psi^*(\mathbf{x}_T)) \sim \mathcal{D}$
4: $\quad \mathcal{L}(\phi, \Xi, \mathbf{x}'_T) = d(\Psi_{\phi,\Xi}(\mathbf{x}'_T), \Psi^*(\mathbf{x}_T))$ subject to $\mathbf{x}'_T \in B(\mathbf{x}_T, r\sigma_T)$
5: $\quad$ Update $\phi$, $\Xi$, and $\mathbf{x}'_T$ using the corresponding gradients $\nabla \mathcal{L}(\phi, \Xi, \mathbf{x}'_T)$
6: $\quad \mathbf{x}'_T \leftarrow \mathbf{x}_T + \mathbf{1}[\|\mathbf{x}'_T - \mathbf{x}_T\|_2 > r] \cdot r \frac{\mathbf{x}'_T - \mathbf{x}_T}{\|\mathbf{x}'_T - \mathbf{x}_T\|_2}$ *// Projected SGD*
7: $\quad$ Update $\mathcal{D}$ with the new $\mathbf{x}'_T$
8: **end while**

---

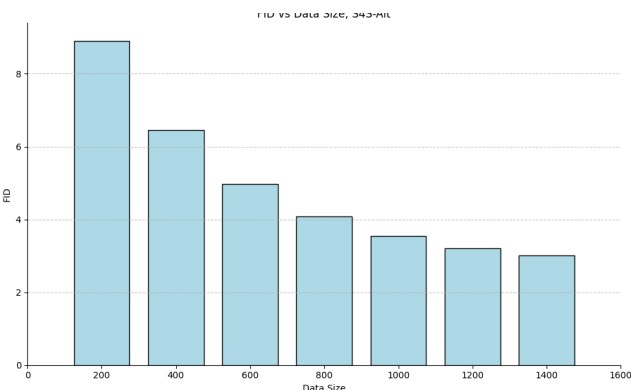

*Figure 4.* FID vs. Training Dataset Size in S4S-Alt.

| Method | CIFAR | | | | MS-COCO | | | |
|---|---|---|---|---|---|---|---|---|
| | NFE | FID | GPU Type | Time | NFE | FID | GPU Type | Time |
| S4S-Alt | 7 | 2.52 | A100 | < 1 hour | 6 | 11.17 | A100 | 4.2 hours |
| S4S | 10 | 2.18 | A100 | < 1 hour | 8 | 10.84 | A100 | 1.4 hours |
| LD3 | 10 | 2.32 | A100 | < 1 hour | 8 | 12.28 | A100 | < 1 hour |
| DPM-v3 | 10 | 2.32 | A40 | 28 hours | 8 | 12.10 | A40 | 88 hours |
| BNS[†] | 8 | 2.73 | - | - | 12 | 20.67 | - | - |
| PD[†] | 8 | 2.57 | TPU | 192 hours | - | - | - | - |
| ECM[†] | 2 | 2.20 | A100 | 192 hours | - | - | - | - |
| iCT-deep[†] | 1 | 2.51 | - | - | - | - | - | - |

*Table 12.* Number of NFEs required to match/beat S4S-Alt performance on CIFAR and MS-COCO. † denotes that results were taken from original papers. PD refers to Progressive Distillation (Salimans & Ho, 2022), ECM to Easy Consistency Models (Geng et al., 2025), iCT-deep to Improved Consistency Training (Song & Dhariwal, 2024). Red cells are methods that cannot match S4S-Alt in our experiments w/ our NFE settings or in recorded experiments.

| Method | Order | NFE=4 | NFE=6 | NFE=8 |
|---|---|---|---|---|
| DPM-Solver-S4S (2S) | 2 | 66.82 | 34.91 | 24.73 |
| Consistent DPM-Solver-S4S (2S) | 2 | 75.82 | 39.14 | 31.69 |

*Table 11.* FID of SS methods initialized at DPM-Solver-S4S on FFHQ with logSNR discretization. Enforcing consistency in the single-step model *decreases* performance rather than achieving better global error.

## H.4. Full FID Tables

| Schedule | Solver | 3 | 4 | 5 | 6 | 7 | 8 | 9 | 10 |
|---|---|---|---|---|---|---|---|---|---|
| DMN | DPM-Solver++ (3M) | 82.45 | 37.52 | 30.08 | 18.40 | 12.31 | 8.95 | 7.40 | 3.69 |
| | DPM-Solver++-S4S (3M) | 75.43 | 34.48 | 28.24 | 17.55 | 11.75 | 8.66 | 7.06 | 3.51 |
| | iPNDM (3M) | 76.99 | 33.13 | 26.10 | 16.00 | 10.20 | 10.19 | 8.84 | 3.56 |
| | iPNDM-S4S (3M) | 69.79 | 30.58 | 24.26 | 15.18 | 9.81 | 9.83 | 8.36 | 3.36 |
| | UniPC (3M) | 70.52 | 30.32 | 23.04 | 14.46 | 8.55 | 6.78 | 5.15 | 3.12 |
| | UniPC-S4S (3M) | **63.84** | **28.43** | **21.66** | **13.88** | **8.24** | **6.53** | **4.84** | **2.98** |
| Time EDM | DPM-Solver++ (3M) | 43.47 | 19.52 | 13.36 | 9.67 | 7.92 | 6.64 | 5.08 | 4.20 |
| | DPM-Solver++-S4S (3M) | 39.90 | 18.32 | 12.55 | 9.11 | 7.61 | 6.37 | 4.86 | 3.96 |
| | iPNDM (3M) | 38.33 | 15.30 | 8.80 | 6.24 | 4.52 | 3.85 | 3.33 | 3.04 |
| | iPNDM-S4S (3M) | **35.56** | **14.23** | **8.32** | **5.97** | **4.37** | **3.77** | **3.12** | **2.88** |
| | UniPC (3M) | 44.77 | 23.55 | 15.83 | 10.30 | 8.46 | 7.83 | 6.78 | 6.38 |
| | UniPC-S4S (3M) | 41.48 | 21.82 | 14.73 | 9.68 | 8.12 | 7.52 | 6.47 | 6.06 |
| GITS | DPM-Solver++ (3M) | 30.74 | 17.73 | 13.57 | 9.91 | 6.99 | 5.31 | 4.26 | 3.62 |
| | DPM-Solver++-S4S (3M) | 28.20 | 16.41 | 12.74 | 9.34 | 6.64 | 5.11 | 4.08 | 3.42 |
| | iPNDM (3M) | 26.55 | 13.88 | 9.60 | 6.10 | 4.85 | 3.72 | 3.43 | 3.02 |
| | iPNDM-S4S (3M) | 24.36 | 12.75 | **9.12** | **5.83** | 4.66 | **3.58** | 3.26 | **2.90** |
| | UniPC (3M) | 25.14 | 12.63 | 9.64 | 7.27 | 4.75 | 4.25 | 3.27 | 3.04 |
| | UniPC-S4S (3M) | **23.36** | **11.56** | 9.13 | 6.85 | **4.55** | 4.08 | **3.13** | 2.95 |
| LD3 | DPM-Solver++ (3M) | 24.11 | 13.95 | 7.46 | 5.66 | 4.00 | 3.61 | 2.75 | 3.04 |
| | DPM-Solver++-S4S (3M) | 21.11 | 12.58 | 6.75 | 5.29 | 3.76 | 3.48 | 2.64 | 2.90 |
| | iPNDM (3M) | 23.64 | 9.06 | 5.00 | 3.44 | 2.78 | 2.87 | 2.85 | 2.62 |
| | iPNDM-S4S (3M) | 20.65 | 8.25 | 4.61 | 3.21 | 2.61 | 2.76 | 2.71 | 2.51 |
| | UniPC (3M) | 22.02 | 10.84 | 6.10 | 3.65 | 3.44 | 3.32 | 2.44 | 2.87 |
| | UniPC-S4S (3M) | 19.38 | 9.69 | 5.61 | 3.40 | 3.27 | 3.19 | 2.32 | 2.69 |
| Time LogSNR | DPM-Solver++ (3M) | 60.83 | 27.58 | 17.92 | 10.72 | 6.14 | 4.31 | 3.63 | 3.15 |
| | DPM-Solver++-S4S (3M) | 55.88 | 25.45 | 16.88 | 10.08 | 5.90 | **4.18** | **3.41** | 2.99 |
| | iPNDM (3M) | 52.63 | 22.99 | 15.58 | 9.45 | 5.92 | 4.51 | 3.71 | 3.14 |
| | iPNDM-S4S (3M) | **48.19** | **21.58** | 14.57 | 8.91 | 5.64 | 4.33 | 3.48 | **2.98** |
| | UniPC (3M) | 94.93 | 33.70 | 12.95 | 8.30 | 5.12 | 4.62 | 4.47 | 3.80 |
| | UniPC-S4S (3M) | 88.13 | 31.23 | **12.18** | **7.91** | **4.85** | 4.47 | 4.26 | 3.62 |
| Time Quadratic | DPM-Solver++ (3M) | 113.09 | 68.88 | 42.36 | 30.99 | 24.82 | 21.04 | 18.66 | 16.93 |
| | DPM-Solver++-S4S (3M) | 103.66 | 63.86 | 39.78 | 29.43 | 23.66 | 20.54 | 17.70 | 16.07 |
| | iPNDM (3M) | 102.48 | 53.71 | 32.09 | 23.86 | 20.36 | 18.22 | 16.62 | 15.23 |
| | iPNDM-S4S (3M) | **94.08** | **49.39** | **29.95** | **22.56** | **19.65** | **17.71** | **15.57** | **14.32** |
| | UniPC (3M) | 111.79 | 66.50 | 41.62 | 30.69 | 24.42 | 20.64 | 18.20 | 16.54 |
| | UniPC-S4S (3M) | 101.64 | 62.03 | 39.30 | 29.17 | 23.63 | 19.89 | 17.05 | 15.77 |
| Time Uniform | DPM-Solver++ (3M) | 169.39 | 153.47 | 143.52 | 134.39 | 125.18 | 115.83 | 106.83 | 98.18 |
| | DPM-Solver++-S4S (3M) | **155.10** | 143.47 | 134.98 | 125.98 | 120.75 | 111.54 | 101.13 | 92.01 |
| | iPNDM (3M) | 178.95 | 159.28 | 139.32 | 124.94 | 113.44 | 102.81 | 92.46 | 82.91 |
| | iPNDM-S4S (3M) | 163.79 | 146.77 | **129.81** | **117.56** | **107.45** | **99.27** | **87.04** | **77.95** |
| | UniPC (3M) | 169.33 | 153.52 | 143.45 | 134.15 | 124.70 | 115.25 | 106.06 | 97.28 |
| | UniPC-S4S (3M) | 156.96 | **142.90** | 135.29 | 127.33 | 120.44 | 111.11 | 99.53 | 91.49 |
| | S4S-Alt | 14.71 | 6.52 | 3.89 | 2.70 | 2.56 | 2.29 | 2.18 | 2.18 |

*Table 13.* FID scores on AFHQ-v2 64×64. Numbers in column headers indicate NFE counts. Bold: best within schedule; shaded: best overall.

| Schedule | Solver | 3 | 4 | 5 | 6 | 7 | 8 | 9 | 10 |
|---|---|---|---|---|---|---|---|---|---|
| DMN | DPM-Solver++ (3M) | 83.73 | 39.32 | 22.89 | 12.38 | 7.23 | 7.00 | 5.20 | 2.69 |
| | DPM-Solver++-S4S (3M) | 70.89 | 34.00 | 20.53 | 11.30 | 6.63 | 6.71 | 4.98 | 2.53 |
| | iPNDM (3M) | 59.31 | 28.08 | 16.76 | 9.24 | 5.77 | 7.59 | 5.85 | 3.17 |
| | iPNDM-S4S (3M) | **50.05** | 24.21 | 14.99 | 8.35 | 5.37 | 7.20 | 5.57 | 3.02 |
| | UniPC (3M) | 66.45 | 26.33 | 12.95 | 8.11 | 4.96 | 5.79 | 4.01 | 2.38 |
| | UniPC-S4S (3M) | 56.44 | 23.07 | **11.63** | 7.44 | **4.66** | 5.53 | 3.79 | **2.26** |
| | DPM-Solver-v3 (3M) | 58.48 | **17.88** | 12.31 | **7.32** | 4.86 | **4.72** | **3.49** | 2.32 |
| Time EDM | DPM-Solver++ (3M) | 70.06 | 50.40 | 32.01 | 18.41 | 11.58 | 8.39 | 6.48 | 5.18 |
| | DPM-Solver++-S4S (3M) | 59.61 | 43.17 | 28.11 | 16.52 | 10.87 | 8.01 | 6.07 | 4.96 |
| | iPNDM (3M) | 48.02 | 29.50 | 16.57 | 9.75 | 6.93 | 5.24 | 4.34 | 3.70 |
| | iPNDM-S4S (3M) | **41.27** | **25.74** | **14.72** | **8.81** | **6.35** | **4.98** | **4.07** | **3.47** |
| | UniPC (3M) | 57.85 | 50.63 | 34.27 | 19.47 | 12.65 | 9.68 | 7.84 | 6.16 |
| | UniPC-S4S (3M) | 48.40 | 44.30 | 30.60 | 17.80 | 11.62 | 9.05 | 7.46 | 5.86 |
| | DPM-Solver-v3 (3M) | 44.64 | 34.39 | 33.20 | 18.44 | 10.50 | 7.39 | 5.91 | 4.72 |
| GITS | DPM-Solver++ (3M) | 70.47 | 31.23 | 17.19 | 10.76 | 7.79 | 5.63 | 3.97 | 3.52 |
| | DPM-Solver++-S4S (3M) | 60.14 | 26.75 | 15.41 | 9.70 | 7.20 | 5.29 | 3.72 | 3.37 |
| | iPNDM (3M) | 43.91 | 16.49 | 10.83 | 6.97 | 5.80 | 4.30 | 3.10 | 2.78 |
| | iPNDM-S4S (3M) | **37.75** | **14.11** | **9.63** | **6.33** | **5.32** | **4.04** | **2.97** | **2.62** |
| | UniPC (3M) | 53.43 | 21.93 | 15.40 | 10.47 | 7.88 | 5.69 | 4.41 | 3.70 |
| | UniPC-S4S (3M) | 45.12 | 18.94 | 13.71 | 9.53 | 7.39 | 5.42 | 4.22 | 3.50 |
| | DPM-Solver-v3 (3M) | 60.14 | 24.46 | 16.15 | 11.06 | 8.20 | 5.90 | 3.88 | 2.99 |
| LD3 | DPM-Solver++ (3M) | 33.38 | 27.08 | 12.42 | 9.24 | 4.40 | 4.00 | 3.87 | 3.33 |
| | DPM-Solver++-S4S (3M) | 27.73 | 22.85 | 10.81 | 8.14 | 4.08 | 3.74 | 3.65 | 3.16 |
| | iPNDM (3M) | 32.64 | 10.93 | 5.64 | 5.40 | 5.36 | 2.75 | 3.79 | 2.32 |
| | iPNDM-S4S (3M) | **26.39** | **9.30** | **4.84** | 4.76 | 4.90 | **2.61** | 3.59 | **2.18** |
| | UniPC (3M) | 32.62 | 15.83 | 13.14 | 3.55 | 4.67 | 2.87 | 3.30 | 2.73 |
| | UniPC-S4S (3M) | 26.63 | 13.46 | 11.35 | **3.17** | 4.22 | 2.67 | 3.09 | 2.56 |
| | DPM-Solver-v3 (3M) | 84.42 | 29.86 | 14.83 | 10.69 | 5.51 | 3.59 | **2.78** | 2.56 |
| Time LogSNR | DPM-Solver++ (3M) | 110.06 | 46.49 | 24.98 | 12.06 | 6.79 | 4.56 | 3.43 | 3.00 |
| | DPM-Solver++-S4S (3M) | 93.58 | 40.18 | 22.21 | 11.04 | 6.34 | 4.34 | 3.25 | 2.85 |
| | iPNDM (3M) | 88.39 | 34.88 | 20.49 | 11.61 | 7.50 | 5.53 | 4.24 | 3.58 |
| | iPNDM-S4S (3M) | **75.88** | 30.12 | 17.97 | **10.61** | 6.91 | 5.24 | 3.99 | 3.43 |
| | UniPC (3M) | 155.31 | 43.93 | 23.90 | 12.98 | 6.54 | 4.38 | 3.48 | 3.07 |
| | UniPC-S4S (3M) | 133.37 | 38.05 | 21.42 | 11.69 | 6.02 | 4.10 | 3.33 | 2.94 |
| | DPM-Solver-v3 (3M) | 84.49 | **29.87** | **14.85** | 10.71 | **5.52** | **3.59** | **2.78** | **2.56** |
| Time Quadratic | DPM-Solver++ (3M) | 223.06 | 170.85 | 124.72 | 91.51 | 69.90 | 54.84 | 44.38 | 37.00 |
| | DPM-Solver++-S4S (3M) | 188.15 | 149.60 | 110.20 | 82.92 | 64.23 | 51.47 | 41.63 | 34.91 |
| | iPNDM (3M) | 199.73 | 139.72 | 96.56 | 68.68 | 52.22 | 37.64 | 27.37 | 23.28 |
| | iPNDM-S4S (3M) | **167.85** | **121.49** | **85.83** | **62.41** | **48.65** | **35.96** | **25.74** | **22.26** |
| | UniPC (3M) | 220.29 | 164.80 | 117.49 | 85.38 | 65.44 | 51.80 | 42.29 | 35.34 |
| | UniPC-S4S (3M) | 187.89 | 141.66 | 103.61 | 78.11 | 61.21 | 49.21 | 39.65 | 33.29 |
| | DPM-Solver-v3 (3M) | 299.55 | 249.40 | 188.77 | 129.51 | 90.93 | 65.13 | 50.30 | 41.07 |
| Time Uniform | DPM-Solver++ (3M) | 305.04 | 282.99 | 263.61 | 249.52 | 237.94 | 227.53 | 217.62 | 208.19 |
| | DPM-Solver++-S4S (3M) | 259.33 | 244.55 | 234.70 | 225.57 | 219.83 | 218.06 | 205.34 | 197.85 |
| | iPNDM (3M) | 287.80 | 266.13 | 242.76 | 229.10 | 216.95 | 205.06 | 194.64 | 185.30 |
| | iPNDM-S4S (3M) | **243.96** | **227.78** | **215.84** | **208.27** | **199.97** | **192.03** | **186.11** | **175.26** |
| | UniPC (3M) | 304.86 | 282.77 | 263.43 | 249.18 | 237.56 | 226.95 | 216.85 | 207.23 |
| | UniPC-S4S (3M) | 255.32 | 246.58 | 235.16 | 226.52 | 222.79 | 214.66 | 203.80 | 194.92 |
| | DPM-Solver-v3 (3M) | 313.89 | 321.04 | 317.36 | 310.77 | 312.46 | 304.90 | 294.57 | 285.39 |
| | S4S-Alt | **16.95** | **6.35** | **3.73** | **2.67** | **2.52** | **2.39** | **2.31** | **2.18** |

*Table 14.* FID scores on CIFAR-10 32×32. Numbers in column headers indicate NFE counts. Bold: best within schedule; shaded: best overall.

| Schedule | Solver | 3 | 4 | 5 | 6 | 7 | 8 | 9 | 10 |
|---|---|---|---|---|---|---|---|---|---|
| DMN | DPM-Solver++ (3M) | 82.21 | 40.23 | 26.30 | 14.74 | 9.78 | 10.10 | 8.63 | 4.63 |
| | DPM-Solver++-S4S (3M) | 68.64 | 34.69 | 23.55 | 13.23 | 9.05 | 9.52 | 8.09 | 4.44 |
| | iPNDM (3M) | 61.76 | 31.28 | 20.93 | 12.12 | 8.62 | 10.95 | 9.81 | 5.29 |
| | iPNDM-S4S (3M) | **52.45** | 27.51 | 18.83 | 11.03 | 8.07 | 10.29 | 9.20 | 5.06 |
| | UniPC (3M) | 65.07 | 25.80 | 13.32 | 9.48 | 7.27 | 6.78 | 5.57 | 3.66 |
| | UniPC-S4S (3M) | 54.58 | **22.32** | **11.79** | **8.57** | **6.69** | **6.39** | **5.30** | **3.50** |
| Time EDM | DPM-Solver++ (3M) | 62.58 | 39.52 | 23.66 | 15.16 | 11.10 | 9.61 | 9.06 | 6.99 |
| | DPM-Solver++-S4S (3M) | 53.21 | 34.47 | 20.93 | 13.75 | 10.20 | 9.07 | 8.52 | 6.62 |
| | iPNDM (3M) | 45.97 | 29.07 | 17.26 | 11.31 | 8.56 | 6.83 | 5.72 | 4.95 |
| | iPNDM-S4S (3M) | **38.45** | **24.90** | **15.10** | **10.37** | **8.01** | **6.42** | **5.42** | **4.75** |
| | UniPC (3M) | 59.88 | 47.73 | 26.54 | 15.07 | 11.20 | 11.65 | 10.91 | 8.89 |
| | UniPC-S4S (3M) | 50.37 | 41.83 | 23.43 | 13.56 | 10.36 | 11.04 | 10.28 | 8.48 |
| GITS | DPM-Solver++ (3M) | 53.42 | 29.07 | 17.54 | 12.74 | 9.74 | 7.70 | 6.30 | 4.99 |
| | DPM-Solver++-S4S (3M) | 45.10 | 25.00 | 15.72 | 11.41 | 9.00 | 7.30 | 5.98 | 4.77 |
| | iPNDM (3M) | 33.09 | 18.04 | 12.91 | 9.38 | 7.57 | 5.76 | 4.76 | 3.97 |
| | iPNDM-S4S (3M) | 28.28 | **15.70** | **11.60** | **8.46** | **7.11** | **5.41** | **4.52** | **3.76** |
| | UniPC (3M) | 43.63 | 21.38 | 14.34 | 12.22 | 9.95 | 8.02 | 6.20 | 4.46 |
| | UniPC-S4S (3M) | 36.59 | 18.30 | 12.68 | 11.19 | 9.20 | 7.56 | 5.83 | 4.24 |
| LD3 | DPM-Solver++ (3M) | 49.86 | 28.67 | 14.39 | 7.70 | 5.01 | 4.21 | 3.56 | 3.41 |
| | DPM-Solver++-S4S (3M) | 41.43 | 23.92 | 12.32 | 6.95 | 4.57 | 3.94 | 3.42 | 3.24 |
| | iPNDM (3M) | 43.05 | 16.68 | 9.41 | 6.19 | 4.62 | 3.75 | 3.41 | 3.13 |
| | iPNDM-S4S (3M) | 35.12 | 14.24 | 8.14 | 5.45 | 4.20 | 3.55 | 3.23 | 2.97 |
| | UniPC (3M) | 40.27 | 18.04 | 10.85 | 8.04 | 4.33 | 3.46 | 3.53 | 3.30 |
| | UniPC-S4S (3M) | **33.33** | 15.09 | 9.34 | 7.09 | 3.91 | 3.23 | 3.34 | 3.12 |
| Time LogSNR | DPM-Solver++ (3M) | 86.39 | 45.89 | 22.52 | 13.78 | 8.47 | 6.06 | 4.77 | 4.12 |
| | DPM-Solver++-S4S (3M) | 74.18 | 39.69 | 19.75 | 12.66 | 7.91 | 5.72 | 4.56 | 3.88 |
| | iPNDM (3M) | 76.81 | 36.23 | 24.16 | 16.15 | 11.07 | 7.93 | 6.27 | 5.30 |
| | iPNDM-S4S (3M) | **65.27** | **31.81** | 21.33 | 14.85 | 10.38 | 7.53 | 5.89 | 5.07 |
| | UniPC (3M) | 126.00 | 53.22 | 20.02 | 10.97 | 6.97 | 5.53 | 4.53 | 3.89 |
| | UniPC-S4S (3M) | 105.40 | 45.53 | **17.73** | **10.09** | **6.48** | **5.19** | **4.28** | **3.72** |
| Time Quadratic | DPM-Solver++ (3M) | 131.14 | 94.28 | 70.33 | 55.02 | 44.74 | 37.45 | 32.26 | 28.46 |
| | DPM-Solver++-S4S (3M) | 112.50 | 82.78 | 61.65 | 49.58 | 41.76 | 35.04 | 30.81 | 26.94 |
| | iPNDM (3M) | 105.90 | 71.59 | 51.72 | 39.21 | 31.40 | 26.52 | 23.42 | 21.30 |
| | iPNDM-S4S (3M) | **90.71** | **62.94** | **45.63** | **35.77** | **28.99** | **25.12** | **22.34** | **20.32** |
| | UniPC (3M) | 128.38 | 89.94 | 66.09 | 51.36 | 41.54 | 34.76 | 29.98 | 26.55 |
| | UniPC-S4S (3M) | 107.65 | 77.61 | 58.90 | 46.69 | 38.75 | 32.51 | 28.48 | 25.04 |
| Time Uniform | DPM-Solver++ (3M) | 195.55 | 179.13 | 165.48 | 153.52 | 142.81 | 133.12 | 124.36 | 116.46 |
| | DPM-Solver++-S4S (3M) | 167.46 | 157.05 | 147.30 | 141.23 | 134.10 | 127.40 | 117.91 | 108.99 |
| | iPNDM (3M) | 177.99 | 160.85 | 146.31 | 133.60 | 122.28 | 112.25 | 103.46 | 95.78 |
| | iPNDM-S4S (3M) | **152.72** | **140.92** | **129.90** | **119.79** | **114.77** | **105.73** | **98.98** | **90.75** |
| | UniPC (3M) | 195.24 | 178.73 | 165.03 | 152.95 | 142.09 | 132.28 | 123.39 | 115.35 |
| | UniPC-S4S (3M) | 164.34 | 154.02 | 146.30 | 138.21 | 133.36 | 125.43 | 117.83 | 109.96 |
| | S4S-Alt | 19.86 | 10.63 | 6.25 | 4.62 | 3.45 | 3.15 | 3.00 | 2.91 |

*Table 15.* FID scores on FFHQ 64×64. Numbers in column headers indicate NFE counts. Bold: best within schedule; shaded: best overall.

| Schedule | Solver | 3 | 4 | 5 | 6 | 7 | 8 |
|---|---|---|---|---|---|---|---|
| | DPM-Solver++ (3M) | 58.17 | 20.03 | 7.14 | 5.04 | 4.69 | 4.53 |
| | DPM-Solver++-S4S (3M) | 55.18 | 19.40 | 6.87 | 4.88 | 4.63 | 4.28 |
| DMN | iPNDM (3M) | 24.73 | 8.15 | 4.74 | 4.38 | 4.51 | 4.42 |
| | iPNDM-S4S (3M) | **23.72** | **7.84** | **4.63** | **4.25** | **4.46** | **4.18** |
| | UniPC (3M) | 48.95 | 15.05 | 5.46 | 4.60 | 4.83 | 4.52 |
| | UniPC-S4S (3M) | 46.99 | 14.36 | 5.33 | 4.43 | 4.80 | 4.31 |
| | DPM-Solver++ (3M) | 123.72 | 60.70 | 18.40 | 7.71 | 5.42 | 5.29 |
| | DPM-Solver++-S4S (3M) | 117.96 | 57.69 | 17.77 | 7.42 | 5.36 | 5.04 |
| Time EDM | iPNDM (3M) | 88.53 | 33.84 | 11.50 | 7.25 | 5.45 | 4.84 |
| | iPNDM-S4S (3M) | **85.46** | **32.50** | **11.21** | 6.99 | 5.36 | **4.53** |
| | UniPC (3M) | 121.64 | 57.32 | 16.05 | 6.92 | 5.45 | 5.47 |
| | UniPC-S4S (3M) | 117.82 | 55.51 | 15.53 | **6.72** | **5.29** | 5.11 |
| | DPM-Solver++ (3M) | 99.24 | 39.63 | 28.21 | 15.79 | 6.98 | 5.20 |
| | DPM-Solver++-S4S (3M) | 95.26 | 37.86 | 27.17 | 15.35 | 6.84 | 4.96 |
| GITS | iPNDM (3M) | 69.12 | 22.22 | 20.73 | 11.79 | 5.64 | 4.51 |
| | iPNDM-S4S (3M) | **66.42** | **21.11** | 20.07 | 11.48 | 5.49 | 4.32 |
| | UniPC (3M) | 85.37 | 24.59 | 16.08 | 8.68 | 4.93 | 4.33 |
| | UniPC-S4S (3M) | 82.13 | 23.61 | **15.42** | **8.53** | **4.79** | **4.11** |
| | DPM-Solver++ (3M) | 52.28 | 17.71 | 6.81 | 4.89 | 4.76 | 4.91 |
| | DPM-Solver++-S4S (3M) | 48.04 | 16.69 | 6.43 | 4.76 | 4.67 | 4.61 |
| LD3 | iPNDM (3M) | 17.93 | 6.45 | 4.86 | 4.70 | 4.73 | 4.91 |
| | iPNDM-S4S (3M) | **16.48** | **6.05** | **4.68** | 4.57 | **4.64** | 4.68 |
| | UniPC (3M) | 43.25 | 11.33 | 5.25 | 4.74 | 4.79 | 4.87 |
| | UniPC-S4S (3M) | 40.24 | 10.56 | 5.05 | **4.54** | 4.71 | **4.58** |
| | DPM-Solver++ (3M) | 111.35 | 55.20 | 14.46 | 6.32 | 5.39 | 5.00 |
| | DPM-Solver++-S4S (3M) | 105.11 | 52.55 | 14.00 | **6.18** | **5.32** | 4.70 |
| Time LogSNR | iPNDM (3M) | 93.77 | 38.81 | 14.79 | 7.70 | 5.61 | 4.85 |
| | iPNDM-S4S (3M) | **89.31** | **36.99** | 14.43 | 7.53 | 5.55 | **4.62** |
| | UniPC (3M) | 109.14 | 50.60 | 12.29 | 6.40 | 5.78 | 5.11 |
| | UniPC-S4S (3M) | 103.63 | 48.95 | **11.94** | 6.18 | 5.67 | 4.89 |
| | DPM-Solver++ (3M) | 91.57 | 40.27 | 17.77 | 8.51 | 5.73 | 4.86 |
| | DPM-Solver++-S4S (3M) | 88.59 | 38.48 | 17.38 | 8.26 | 5.65 | 4.65 |
| Time Quadratic | iPNDM (3M) | 63.67 | 22.65 | 11.32 | 6.60 | 5.02 | 4.48 |
| | iPNDM-S4S (3M) | **60.36** | **21.87** | 10.98 | 6.39 | 4.91 | 4.28 |
| | UniPC (3M) | 82.66 | 28.84 | 11.06 | 5.73 | 4.65 | 4.37 |
| | UniPC-S4S (3M) | 78.47 | 27.66 | **10.77** | **5.61** | **4.62** | **4.09** |
| | DPM-Solver++ (3M) | 68.92 | 26.34 | 9.95 | 6.12 | 5.27 | 5.06 |
| | DPM-Solver++-S4S (3M) | 65.90 | 25.21 | 9.71 | 5.99 | 5.16 | 4.81 |
| Time Uniform | iPNDM (3M) | 32.79 | 8.63 | 5.23 | 4.67 | 4.66 | 4.69 |
| | iPNDM-S4S (3M) | **31.58** | **8.29** | **5.13** | **4.52** | **4.61** | **4.50** |
| | UniPC (3M) | 63.78 | 20.14 | 7.58 | 5.34 | 5.06 | 5.02 |
| | UniPC-S4S (3M) | 61.41 | 19.50 | 7.27 | 5.20 | 4.91 | 4.80 |
| | S4S-Alt | **13.26** | **5.13** | **4.30** | **4.09** | **4.06** | **4.06** |

*Table 16.* FID scores on ImageNet 256×256. Numbers in column headers indicate NFE counts. Bold: best within schedule; shaded: best overall.

| Schedule | Solver | 3 | 4 | 5 | 6 | 7 | 8 |
|---|---|---|---|---|---|---|---|
| DMN | DPM-Solver++ (3M) | 136.53 | 79.29 | 39.67 | 22.66 | 17.18 | 15.19 |
| | DPM-Solver++-S4S (3M) | 116.46 | 71.36 | 35.82 | 21.42 | 16.56 | 14.48 |
| | iPNDM (3M) | 76.56 | 45.73 | 30.09 | 20.57 | 18.02 | 18.70 |
| | iPNDM-S4S (3M) | **66.77** | **40.05** | 27.40 | 19.24 | 17.17 | 17.70 |
| | UniPC (3M) | 126.55 | 68.04 | 31.75 | 19.46 | 15.27 | 14.81 |
| | UniPC-S4S (3M) | 110.11 | 60.68 | 29.18 | 18.40 | 14.57 | 13.90 |
| | DPM-Solver-v3 (3M) | 96.83 | 42.95 | **19.82** | **12.81** | **11.10** | **12.59** |
| Time EDM | DPM-Solver++ (3M) | 213.95 | 141.37 | 80.75 | 46.62 | 30.78 | 22.73 |
| | DPM-Solver++-S4S (3M) | 186.70 | 126.09 | 73.06 | 43.20 | 29.55 | 21.57 |
| | iPNDM (3M) | 126.55 | 88.04 | 52.94 | 37.81 | 31.41 | 26.39 |
| | iPNDM-S4S (3M) | **109.90** | **77.86** | 48.79 | 35.27 | 30.26 | 24.78 |
| | UniPC (3M) | 211.69 | 135.41 | 75.79 | 44.78 | 30.13 | 22.20 |
| | UniPC-S4S (3M) | 183.16 | 121.59 | 70.02 | 42.48 | 28.73 | 21.10 |
| | DPM-Solver-v3 (3M) | 159.55 | 114.26 | **43.56** | **23.64** | **19.59** | **15.68** |
| GITS | DPM-Solver++ (3M) | 169.26 | 121.24 | 90.64 | 72.89 | 64.37 | 61.41 |
| | DPM-Solver++-S4S (3M) | 148.05 | 108.86 | 83.25 | 67.94 | 61.69 | 58.83 |
| | iPNDM (3M) | 126.10 | 109.43 | 94.83 | 75.52 | 64.55 | 59.71 |
| | iPNDM-S4S (3M) | **108.68** | 97.21 | 85.43 | 71.03 | 61.71 | 56.04 |
| | UniPC (3M) | 150.08 | 103.39 | 86.62 | 71.78 | 61.22 | 56.08 |
| | UniPC-S4S (3M) | 129.69 | **91.15** | **79.82** | 67.24 | 59.50 | 53.15 |
| | DPM-Solver-v3 (3M) | 166.97 | 129.41 | 92.66 | **64.21** | **50.95** | **47.19** |
| LD3 | DPM-Solver++ (3M) | 144.81 | 77.15 | 42.05 | 23.09 | 15.62 | 12.45 |
| | DPM-Solver++-S4S (3M) | 121.66 | 66.97 | 38.19 | 21.57 | 15.11 | 11.72 |
| | iPNDM (3M) | 73.94 | 35.58 | 20.55 | 14.99 | 12.37 | 11.45 |
| | iPNDM-S4S (3M) | **61.83** | **30.82** | **18.69** | **13.84** | 11.76 | 10.85 |
| | UniPC (3M) | 130.65 | 59.89 | 31.89 | 17.33 | 13.33 | 10.53 |
| | UniPC-S4S (3M) | 110.02 | 52.03 | 28.69 | 16.29 | 12.75 | 9.92 |
| | DPM-Solver-v3 (3M) | 110.47 | 52.81 | 23.85 | 15.08 | **11.11** | **9.73** |
| Time LogSNR | DPM-Solver++ (3M) | 227.26 | 113.01 | 63.69 | 41.60 | 32.96 | 27.31 |
| | DPM-Solver++-S4S (3M) | 198.27 | 101.50 | 57.33 | 39.04 | 31.62 | 25.61 |
| | iPNDM (3M) | 192.47 | 96.55 | 62.97 | 45.45 | 35.94 | 29.28 |
| | iPNDM-S4S (3M) | **167.90** | **85.62** | 57.38 | 42.41 | 34.80 | 27.86 |
| | UniPC (3M) | 223.84 | 106.68 | 62.32 | 43.15 | 34.15 | 26.99 |
| | UniPC-S4S (3M) | 195.09 | 95.94 | 57.50 | 40.87 | 33.21 | 25.52 |
| | DPM-Solver-v3 (3M) | 193.57 | 86.05 | **36.58** | **22.02** | **19.16** | **17.16** |
| Time Quadratic | DPM-Solver++ (3M) | 159.24 | 93.01 | 56.11 | 39.58 | 30.28 | 26.43 |
| | DPM-Solver++-S4S (3M) | 139.02 | 82.74 | 51.65 | 37.37 | 29.10 | 25.35 |
| | iPNDM (3M) | 110.97 | 71.14 | 52.42 | 42.89 | 34.06 | 28.76 |
| | iPNDM-S4S (3M) | **95.17** | **63.24** | 48.46 | 40.36 | 33.04 | 27.18 |
| | UniPC (3M) | 147.85 | 78.28 | 47.40 | 36.08 | 28.10 | 24.16 |
| | UniPC-S4S (3M) | 127.10 | 70.17 | 43.36 | 33.58 | 27.28 | 22.69 |
| | DPM-Solver-v3 (3M) | 136.22 | 64.17 | **38.96** | **25.96** | **23.31** | **20.50** |
| Time Uniform | DPM-Solver++ (3M) | 155.60 | 84.95 | 39.63 | 22.84 | 15.36 | 12.25 |
| | DPM-Solver++-S4S (3M) | 134.61 | 75.20 | 36.64 | 21.28 | 14.86 | 11.55 |
| | iPNDM (3M) | 86.76 | 34.61 | 19.56 | 15.85 | 13.52 | 12.17 |
| | iPNDM-S4S (3M) | **75.52** | **30.95** | **17.69** | **14.99** | 12.95 | 11.40 |
| | UniPC (3M) | 150.76 | 73.74 | 31.62 | 18.22 | 12.66 | 10.31 |
| | UniPC-S4S (3M) | 131.51 | 65.59 | 28.62 | 17.05 | 12.20 | **9.69** |
| | DPM-Solver-v3 (3M) | 110.45 | 52.81 | 23.85 | 15.08 | **11.12** | 9.73 |
| | S4S-Alt | **37.65** | **20.89** | **13.03** | **10.49** | **10.03** | **9.64** |

*Table 17.* FID scores on LSUN-Bedroom 256×256. Numbers in column headers indicate NFE counts. Bold: best within schedule; shaded: best overall. Curiously, despite using essentially the same replication code as in Zheng et al. (2023) and Tong et al. (2024) for LSUN-Bedroom generation, we were persistently unable to achieve the FID stated in many papers; accordingly, we present this mainly as demonstrating the overall trend for S4S on LSUN-Bedroom.

| Schedule | Solver | 3 | 4 | 5 | 6 | 7 | 8 |
|---|---|---|---|---|---|---|---|
| DMN | DPM-Solver++ (3M) | 65.43 | 26.54 | 20.40 | 15.36 | 14.42 | 13.14 |
| | DPM-Solver++-S4S (3M) | 60.26 | 24.81 | 19.50 | 14.92 | 13.96 | 12.44 |
| | iPNDM (3M) | 66.77 | 27.16 | 19.42 | 14.26 | 11.74 | 11.85 |
| | iPNDM-S4S (3M) | 61.37 | 25.45 | 18.68 | **13.88** | **11.40** | **11.13** |
| | UniPC (3M) | 60.75 | 24.18 | 18.21 | 14.59 | 14.26 | 14.03 |
| | UniPC-S4S (3M) | **56.42** | **22.47** | **17.51** | 14.12 | 13.97 | 13.24 |
| | DPM-Solver-v3 (3M) | 107.07 | 59.43 | 49.97 | 30.55 | 23.10 | 18.98 |
| Time EDM | DPM-Solver++ (3M) | 61.90 | 33.61 | 40.91 | 28.52 | 16.72 | 12.35 |
| | DPM-Solver++-S4S (3M) | 57.01 | 31.59 | 39.10 | 27.51 | 16.46 | 11.63 |
| | iPNDM (3M) | 46.66 | 28.48 | 19.66 | 14.77 | 12.28 | 11.45 |
| | iPNDM-S4S (3M) | **43.48** | **26.59** | **18.90** | **14.05** | **11.87** | **10.72** |
| | UniPC (3M) | 62.90 | 36.93 | 53.93 | 49.19 | 33.00 | 19.59 |
| | UniPC-S4S (3M) | 57.67 | 34.68 | 51.46 | 46.84 | 32.40 | 18.54 |
| | DPM-Solver-v3 (3M) | 97.31 | 64.57 | 77.71 | 65.88 | 34.62 | 17.82 |
| GITS | DPM-Solver++ (3M) | 39.62 | 20.57 | 19.18 | 13.64 | 12.52 | 11.72 |
| | DPM-Solver++-S4S (3M) | **36.87** | 19.53 | 18.04 | 13.03 | 12.23 | **11.00** |
| | iPNDM (3M) | 43.06 | 23.29 | 16.40 | 12.33 | 11.56 | 11.36 |
| | iPNDM-S4S (3M) | 39.54 | 21.78 | **15.69** | **11.97** | **11.23** | 10.82 |
| | UniPC (3M) | 39.42 | 20.22 | 22.25 | 14.63 | 12.40 | 11.30 |
| | UniPC-S4S (3M) | 36.94 | **19.14** | 21.28 | 13.96 | 12.08 | 11.14 |
| | DPM-Solver-v3 (3M) | 70.07 | 35.45 | 27.86 | 15.31 | 13.31 | 12.10 |
| LD3 | DPM-Solver++ (3M) | 34.32 | 20.64 | 15.47 | 14.26 | 14.07 | 13.67 |
| | DPM-Solver++-S4S (3M) | 31.77 | **19.21** | **14.83** | 13.89 | 13.75 | 13.01 |
| | iPNDM (3M) | 43.73 | 26.14 | 17.33 | 13.19 | 12.31 | 12.28 |
| | iPNDM-S4S (3M) | 40.75 | 24.60 | 16.27 | **12.63** | **11.89** | **11.69** |
| | UniPC (3M) | 33.94 | 21.27 | 16.27 | 14.55 | 14.49 | 13.03 |
| | UniPC-S4S (3M) | **31.03** | 19.70 | 15.30 | 13.95 | 14.23 | 12.26 |
| | DPM-Solver-v3 (3M) | 49.07 | 23.92 | 17.35 | 15.22 | 14.43 | 14.11 |
| Time LogSNR | DPM-Solver++ (3M) | 61.46 | 36.02 | 27.02 | 19.31 | 13.86 | 11.76 |
| | DPM-Solver++-S4S (3M) | 56.23 | 33.93 | 25.49 | 18.56 | 13.48 | 11.16 |
| | iPNDM (3M) | 52.27 | 30.47 | 20.11 | 15.18 | 12.65 | 11.60 |
| | iPNDM-S4S (3M) | **48.28** | **28.88** | **18.98** | **14.71** | **12.48** | **10.94** |
| | UniPC (3M) | 61.08 | 37.31 | 32.95 | 27.20 | 19.12 | 14.30 |
| | UniPC-S4S (3M) | 57.23 | 34.56 | 31.41 | 26.11 | 18.58 | 13.73 |
| | DPM-Solver-v3 (3M) | 99.16 | 64.39 | 44.02 | 33.51 | 24.23 | 15.99 |
| Time Quadratic | DPM-Solver++ (3M) | 63.33 | 28.33 | 17.00 | 13.57 | 12.34 | 11.82 |
| | DPM-Solver++-S4S (3M) | 58.16 | 26.61 | 16.25 | 13.21 | 11.89 | 11.35 |
| | iPNDM (3M) | 59.94 | 27.93 | 16.65 | 13.03 | 11.84 | 11.48 |
| | iPNDM-S4S (3M) | **54.75** | 26.19 | 15.76 | **12.42** | **11.67** | **10.84** |
| | UniPC (3M) | 60.51 | 26.47 | 16.49 | 13.41 | 12.26 | 11.74 |
| | UniPC-S4S (3M) | 56.50 | **24.95** | **15.67** | 13.05 | 11.87 | 11.07 |
| | DPM-Solver-v3 (3M) | 103.62 | 60.78 | 34.99 | 21.88 | 16.17 | 13.42 |
| Time Uniform | DPM-Solver++ (3M) | 34.57 | 21.24 | 17.09 | 15.54 | 14.82 | 14.50 |
| | DPM-Solver++-S4S (3M) | **31.67** | **20.05** | **16.03** | 14.85 | 14.57 | 13.68 |
| | iPNDM (3M) | 48.29 | 28.75 | 18.52 | 14.40 | 13.00 | 12.78 |
| | iPNDM-S4S (3M) | 44.89 | 27.27 | 17.45 | **14.00** | **12.77** | **12.11** |
| | UniPC (3M) | 35.33 | 21.42 | 17.31 | 15.39 | 14.65 | 14.36 |
| | UniPC-S4S (3M) | 33.04 | 20.32 | 16.57 | 14.83 | 14.30 | 13.78 |
| | DPM-Solver-v3 (3M) | 49.07 | 23.92 | 17.37 | 15.22 | 14.43 | 14.11 |
| | S4S-Alt | **25.44** | **16.05** | **13.26** | **11.17** | **10.83** | **10.68** |

*Table 18.* FID scores on MS-COCO 512×512. Numbers in column headers indicate NFE counts. Bold: best within schedule; shaded: best overall.

## H.5. Qualitative Model Samples

We provide qualitative samples below.

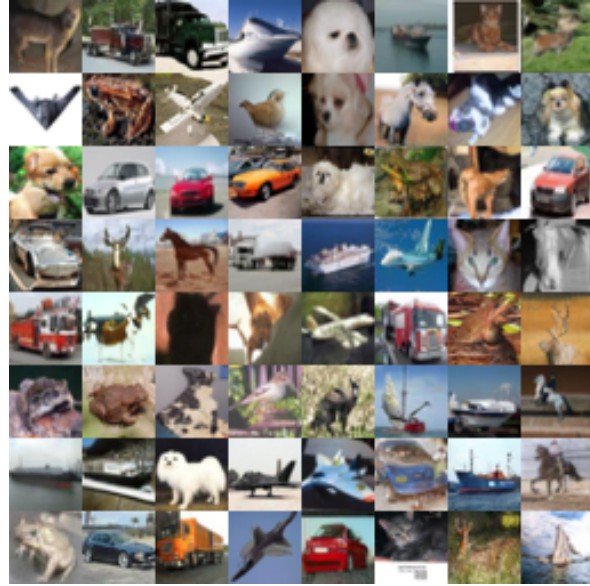

(a) Teacher Solver – UniPC, 20NFE, logSNR, 2.03 FID

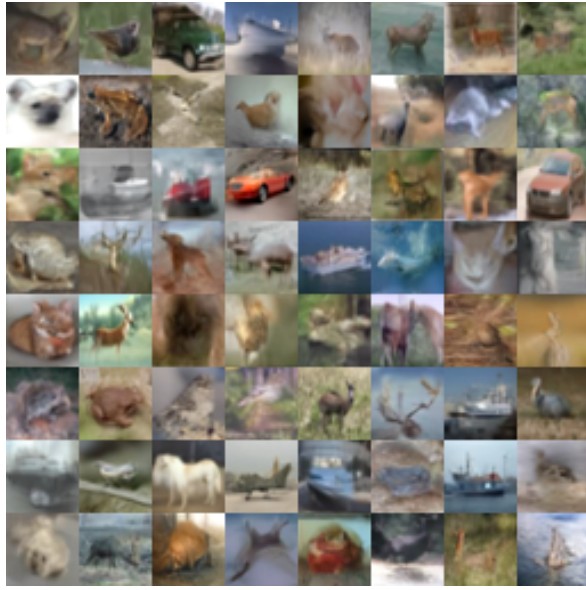

(b) iPNDM, 5 NFE, logSNR, 20.49 FID

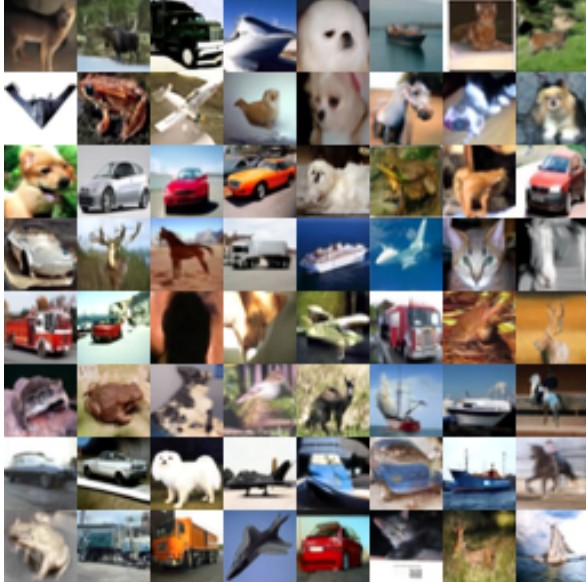

(c) iPNDM-S4S, 5 NFE, LD3, 4.84 FID

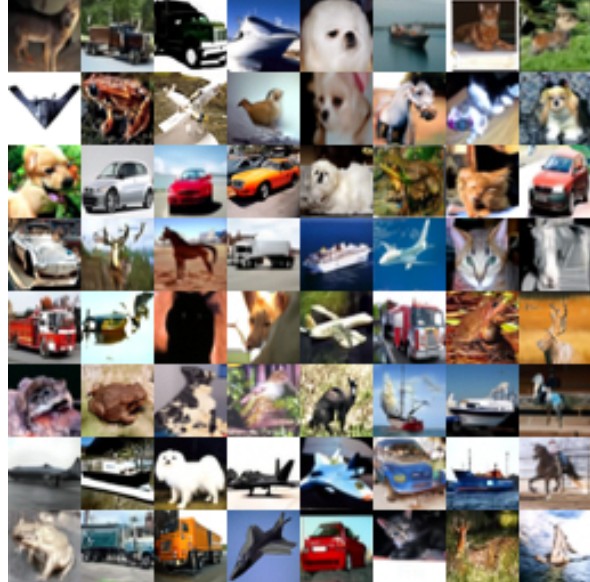

(d) S4S-Alt, 5 NFE, 3.73 NFE

*Figure 5.* Examples from CIFAR-10 32×32

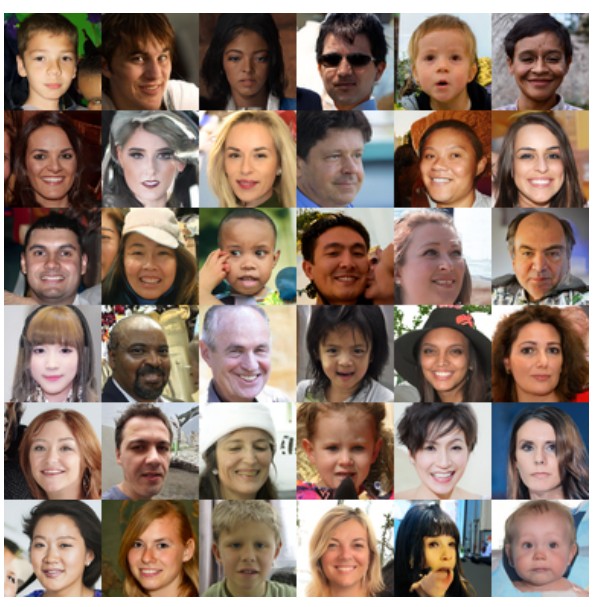

(a) Teacher Solver – UniPC, 20NFE, logSNR, 2.62 FID

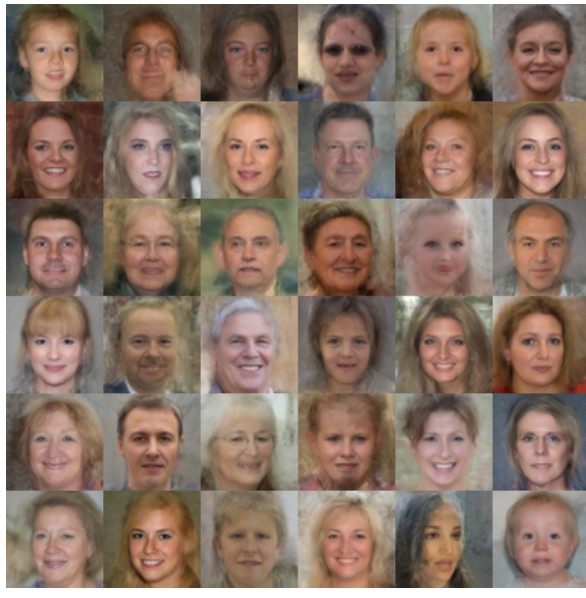

(b) iPNDM, 6 NFE, logSNR, 16.15 FID

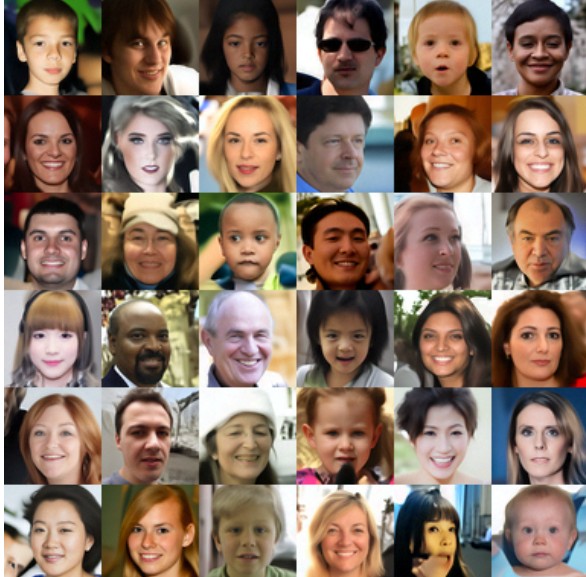

(c) S4S, 6 NFE, LD3, 5.45 FID

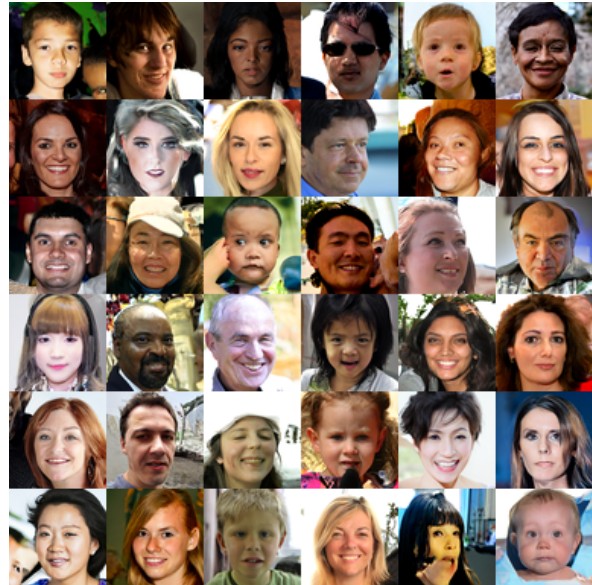

(d) S4S-Alt, 6 NFE, 4.62 FID

*Figure 6.* Examples from FFHQ 64×64

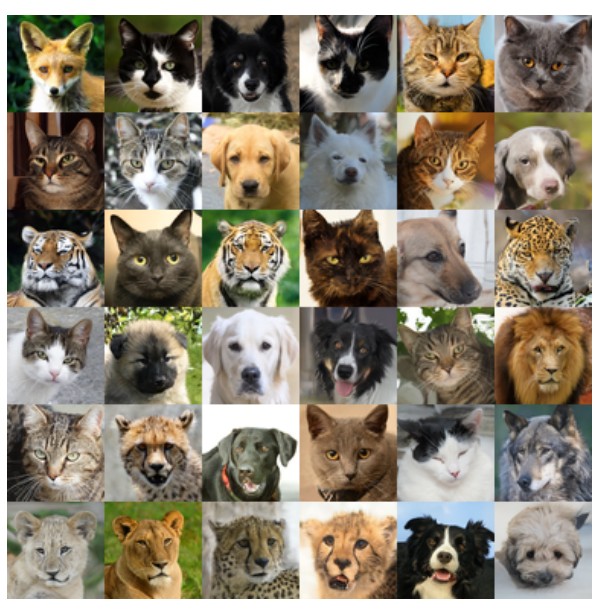

(a) Teacher Solver – UniPC, 20NFE, logSNR, 2.24 FID

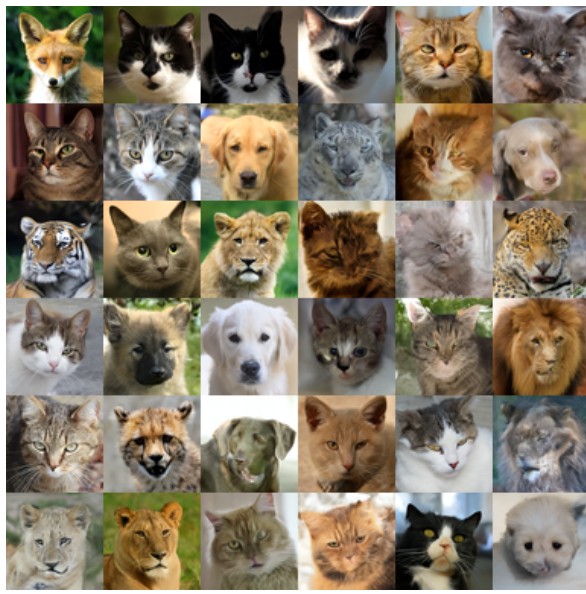

(b) iPNDM, 7 NFE, logSNR, 5.92 FID

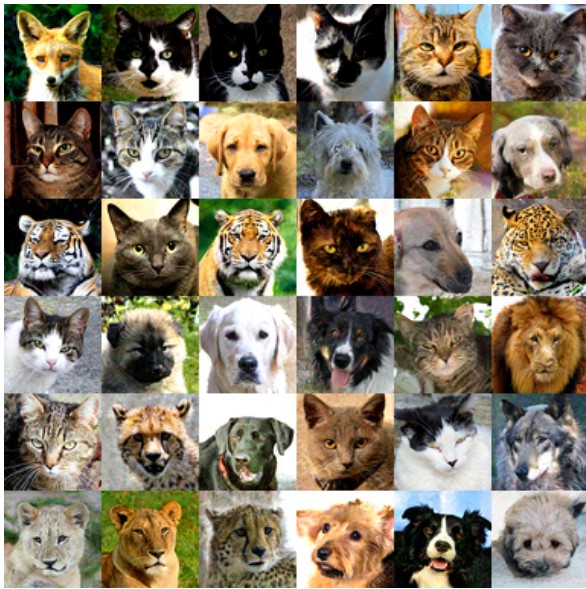

(c) S4S, 7 NFE, LD3, 2.61 FID

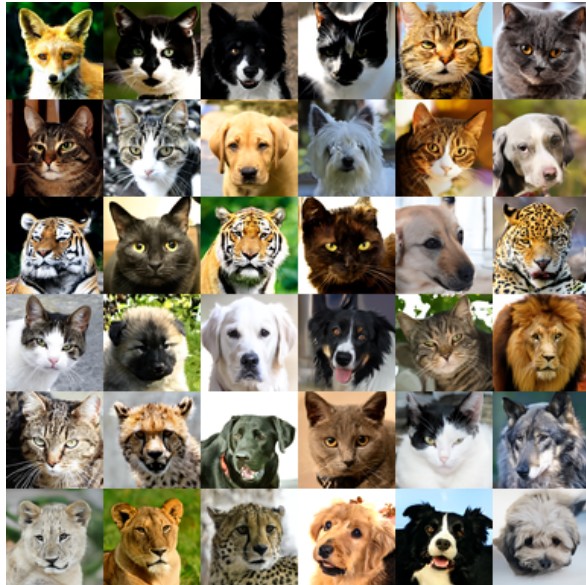

(d) S4S-Alt, 7 NFE, 2.56 FID

*Figure 7.* Examples from AFHQv2 64×64

