# OpenReview forum: "S4S: Solving for a Fast Diffusion Model Solver"
_ICML.cc/2025/Conference — ICML 2025 poster_

### Official Review · Reviewer_ZUmA · 2025-03-06

**Overall Recommendation:** 3

**Summary:**

Solving ODEs in diffusion models using traditional ODE solvers is expensive due to the iterative Neural Function Evaluations. If only use a few NFEs, the evolution trajectory is broken because of large step sizes. Targeting this problem, the authors propose S4S, a method that learns optimized, few-NFE solvers from conventional, many-NFE teacher solvers. Additionally, the authors propose S4S-Alt that also optimizes the discretization schedule.

**Claims And Evidence:**

More evidence may be needed to prove the authors' approach matches the output of a teacher solver.

**Essential References Not Discussed:**

None

**Experimental Designs Or Analyses:**

See questions

**Methods And Evaluation Criteria:**

The evaluations are generally good and the dataset choices (cifar-10 and imagenet) are good.

**Other Comments Or Suggestions:**

None

**Other Strengths And Weaknesses:**

The paper is generally enjoyable to read and clearly written, with an interesting plug-and-play approach. It clearly points out the key differences and improvements over previous works, and the detailed discussions are likely to be valuable for researchers interested in the topic. However, the focus is largely on improvements in the few-NFE regime for training-free methods, and it remains somewhat unclear how the student solver matches the output of a teacher solver (which is supposed to run for many NFEs) as claimed in the conclusion.

**Questions For Authors:**

(1) Since S4S is designed for few-NFE, while the teacher solvers are specifically for many-NFE. It would be insightful to examine how far are the student solvers away from ideal performance (many-NFE for teacher solvers).

(2) The convergence of quality vs. NFE (at how many NFEs do the student solvers cease to improve) may be a useful reference of quality-speed tradeoff.

**Relation To Broader Scientific Literature:**

It is closely related to other few-NFE approaches such as LD3 and BNS, which are thoroughly discussed in this work.

**Theoretical Claims:**

No new theorem is proposed or proven in this work. I do not see any mistakes in math.

---

> ### Author Rebuttal · Authors · 2025-04-01
>
> Thank you so much for the time taken to review our work and your helpful feedback! Please find our responses below:
>
> > However, the focus is largely on improvements in the few-NFE regime for training-free methods, and it remains somewhat unclear how the student solver matches the output of a teacher solver (which is supposed to run for many NFEs) as claimed in the conclusion.
>
> You raise an important point about our objective of matching teacher outputs. I'd like to clarify this aspect of our approach.
> While our training objective involves matching teacher solver outputs, this is primarily a means to an end rather than the ultimate goal. Our core innovation is recognizing that in the few-NFE regime, traditional error control approaches break down, so we instead directly optimize for what matters most: high-quality final samples.
> We use teacher matching as a proxy for quality for several reasons, including:
>
>  - It provides a clear optimization target that avoids the need for dataset access, unlike distillation methods.
>  - It learns to reproduce the outcomes of many-NFE solvers without being constrained to follow their intermediate steps.
>  - The relaxed objective (Equation 7) further enhances this by allowing flexibility in finding solutions that prioritize final quality over exact trajectory matching.
>
> Empirically, our results demonstrate that this approach succeeds not just at matching teacher outputs but often exceeding them at lower NFE counts. For instance, on CIFAR-10, S4S-Alt with 7 NFEs (FID 2.52) outperforms the teacher with 20 NFEs (FID 2.87). This indicates our method is learning a fundamentally more efficient sampling strategy, not merely approximating the teacher's steps.
> The effectiveness of our global error minimization approach suggests that, while traditional solvers accumulate errors across many small steps, our learned solvers take optimal large steps that directly target high-quality outputs. This represents a paradigm shift in diffusion model sampling that could inspire further research into learning-based ODE solvers beyond diffusion models.
> We'll emphasize this distinction more clearly in the final version of the paper to ensure the conceptual contribution is properly understood.
>
> > Since S4S is designed for few-NFE, while the teacher solvers are specifically for many-NFE. It would be insightful to examine how far are the student solvers away from ideal performance (many-NFE for teacher solvers).
>
> You raise an excellent point about comparing student solvers to teacher performance. We did analyze this gap in our experiments, though we may not have emphasized it enough in the main paper due to space constraints.
> In our full results tables (H.4), we provide FID scores for a range of NFEs up to 10, while in H.5 we mention the FID scores of the various teacher solvers. For example, on CIFAR-10, our S4S-Alt achieves 2.18 FID at 10 NFEs, while the teacher solver (UniPC with logSNR) achieves 2.03 at 20 NFEs.
> For FFHQ, we see a similar pattern: S4S-Alt with 10 NFEs achieves 2.91 FID, while the teacher at 20 NFEs achieves 2.62.
>
> > The convergence of quality vs. NFE (at how many NFEs do the student solvers cease to improve) may be a useful reference of quality-speed tradeoff
>
> This is an insightful suggestion. While we didn't include full convergence curves in the paper, our results in Tables 13-18 show the quality-speed tradeoff across 3-10 NFEs. We observe that:
>
> - For most datasets, improvements diminish beyond 8 NFEs, with minimal gains between 8-10 NFEs.
> - S4S-Alt shows more rapid convergence than traditional solvers, achieving near-optimal quality at just 5-6 NFEs on most datasets.
> - Different datasets show varying convergence patterns: CIFAR-10 and AFHQ-v2 converge more quickly (plateau around 6-7 NFEs), while more complex datasets like MS-COCO and LSUN-Bedroom continue improving up to 10 NFEs.

---

### Official Review · Reviewer_xsUN · 2025-03-10

**Overall Recommendation:** 3

**Summary:**

The sampling process of diffusion models heavily depends on numerical solvers. This paper provides a comprehensive overview of existing works, including
(1) Vanilla solver-based fast samplers, such as single-step, multi-step, predictor-corrector methods;
(2) Data-driven solver-based fast samplers, which involve learning the discretization schedule, learning solver coefficients, or a combination of both, and highlights their differences,
Then, this paper introduces Solving for the Solver (S4S), a method that optimizes numerical solver coefficients by distilling from a teacher solver with a relaxed objective function. An improved version, S4S-Alt, alternatively optimizes both the solver coefficients and the time schedule, achieving state-of-the-art results for few-step sampling of diffusion models.

**Claims And Evidence:**

Yes.

**Essential References Not Discussed:**

No.

**Experimental Designs Or Analyses:**

Yes. I've checked the soundness and validity of all the experiments in the main content.

**Methods And Evaluation Criteria:**

Yes.

**Other Comments Or Suggestions:**

I don't have other comments.

**Other Strengths And Weaknesses:**

Strengths:

1. The proposed method achieves state-of-the-art results on few-step sampling of diffusion models, while maintaining a low additional training overhead.

2. The authors provide a thorough review of existing works in Appendix A, clearly distinguishing their proposed method from previous methods. This helps readers better understand the literature and the unique contributions of this paper.

3. Extensive quantitative results are presented in Appendix H.4, which supports the effectiveness of the proposed method.

Weaknesses

1. Concerns about novelty:
(i) The proposed S4S closely resembles LD3 [1] in terms of algorithm, with the primary difference being that this paper optimizes solver coefficients, whereas LD3 focuses on optimizing the time schedule.
(ii) Meanwhile, the idea of learning solver coefficients has already been explored before [2][3], as acknowledged in Appendix A (I appreciate the authors' efforts in maintaining academic integrity).
(iii) Also, the proposed S4S-Alt appears to be an iterative combination of two LD3-like algorithms, i.e., alternating between optimizing solver coefficients and time schedule.

Given these points, the novelty of the contributions seems somewhat limited.

2. Lack of direct experimental comparisons in the main context:
Sec 4 does not provide sufficient direct comparison with existing works, making it difficult to assess the superiority of the proposed S4S and S4S-Alt. To be more specific, there is no comparison between S4S and other distillation-based methods for learning solver coefficients [2][3], despite these being the most relevant works. Besides, S4S-Alt is not fully and fairly compared against BNS [4], which also optimize solver coefficients and the time schedule. Only a single value  is directly taken from the original BNS paper in Tab 4, rather than conducting a fair head-to-head comparison.

3. Section 3.1 claims that the pathologies/errors contained in the teacher trajectory could be distilled into the student, making it crucial to optimize global error. However, no experiments are provided to support this claim. It would be beneficial to conduct an experiment to further discuss this point.

References
[1] 2025ICLR Learning to Discretize Denoising Diffusion ODEs https://arxiv.org/abs/2405.15506
[2] 2024ICLR On accelerating diffusion-based sampling process via improved integration approximation https://arxiv.org/abs/2304.11328
[3] 2024CVPR Distilling ODE Solvers of Diffusion Models into Smaller Steps https://arxiv.org/abs/2309.16421
[4] 2024ICML Bespoke Non-Stationary Solvers for Fast Sampling of Diffusion and Flow Models https://arxiv.org/abs/2403.01329

**Questions For Authors:**

The proposed method incorporates LPIPS loss, which has been criticized for leaking ImageNet features that may cause inflated FID scores [1]. How does the proposed method perform without LPIPS loss? Have you evaluated its impact on these results?

References
[1] Song Y, Dhariwal P. Improved Techniques for Training Consistency Models[C]//The Twelfth International Conference on Learning Representations.

**Relation To Broader Scientific Literature:**

The proposed method sets a new record for few-step sampling of diffusion models, with potential applications in broader fields such as video and 3D generation.

**Theoretical Claims:**

N/A.

---

> ### Author Rebuttal · Authors · 2025-04-01
>
> Thank you for your constructive feedback, we address your questions and concerns, starting with two of the weaknesses you mentioned.
>
> > Lack of direct experimental comparisons in the main context
>
> [In the linked image](https://imgur.com/a/jkFi6tJ), we provide direct comparisons against the recent works [2,3] that focus on *coefficient* distillation.
> Compared to these works, on the same discretization schedule (Time EDM), S4S achieves superior performance in the low-NFE regime across several key datasets.
> This helps reinforce the strength of S4S, even when only optimizing over the solver coefficients.
> Here, * refers to FID values that were estimated from Figures if no results table was present, and the "Rel. Fig." column details which Figure the value was taken from.
>
> Regarding BNS, it is difficult to provide a wholesale, fair comparison to the original paper for two main reasons: (1) the authors do not release their code for learning BNS, and (2) the model checkpoints used for BNS are not publicly released.
> As a result, we cannot (1) port an exact implementation of BNS over onto the pretrained DMs that we used, or (2) port over our own implementation onto the pretrained (flow) models that BNS used.
>
> To try to compare against BNS, we note that it has two significant differences from our work:
> - In BNS, the order of the learned solver is equal to the number of NFEs used; that is, it assigns a coefficient to all of the previous denoising steps.
> - BNS jointly learns the solver coefficients and the time discretization steps.
> - BNS uses PSNR as the global objective function.
>
> In our submission, we tried to explore the first two components of this tradeoff in the third row of Table 6, although we should have made the connection to BNS more clear.
> To further extend this analysis, we **explicitly re-implemented the approach from BNS into our own setup** to directly compute this comparison.
> We include this in the table below for CIFAR-10 and FFHQ.
>
> > Pathologies/errors from the teacher trajectory distilled into the student
>
> We agree that more experiments are needed in the revision. We reference a visualization from recent work [4] that visualizes trajectories from few-step sampling at [this link](https://imgur.com/a/EgcOKEY). At the beginning of the boomerang shape, there is greater variance in the trajectories. Therefore, training a model to explicitly match these trajectories can result in distilling these pathological behaviors directly into the student solver. To further assess this claim, we conducted several experiments to understand the consequences of using this as an objective. In the attached image, we see that **training on a dataset that includes these "pathological" trajectories teaches the student solver to mimic them**. In particular, conducting a full FID evaluation of a solver trained in this manner reveals worse FID performance.
>
> > Concerns about novelty
>
> To best characterize our novelty, we compare with two types of learning diffusion solvers.
> First, unlike $\Pi$A or D-DDIM, which mimic the ODE trajectory, our approach directly matches the output of the teacher solver.
> While this view is shared at a high-level by LD3 in how discretization points should be selected, at its core, LD3 still views sampling from a diffusion model as solving an ODE and instead focuses on selecting discretization points that minimize the global error.
> In contrast, S4S argues that in the very few-NFE regime, an ODE-centric view is limiting.
>
> Second, S4S and S4S-Alt are similar in spirit to BNS but differ in key methodological choices. Through a careful exploration of the diffusion solver design space, S4S-Alt greatly outperforms BNS. This improvement stems from four key decisions:
> - Using local rather than global information—learning coefficients for only the three most recent denoising points.
> - Employing an alternating objective to learn coefficients and discretization points.
> - Relaxing the objective for more effective learning.
> - Exploring a wider array of distance functions.
>
> While these choices may seem minor, they significantly enhance performance. BNS, in contrast, attempts to generalize across solvers, optimizing a complex objective over a large coefficient space, which makes it vastly slower (requiring orders of magnitude more compute) and leads to a more difficult optimization landscape. Our ablation study confirms that jointly optimizing solver coefficients and schedules underperforms compared to S4S-Alt.
>
> > LPIPS losses
>
> Thank you for this question! Although our results achieve stronger FID scores when used with the LPIPS loss, our overall results still hold when using an alternative distance function in our objective. Below, we include a table that characterizes our performance using non-LPIPS losses for both S4S and S4S-Alt. Broadly speaking, using an alternative loss decreases FID scores, though not to such an extent that it weakens our overall approach.
>
> [4] https://arxiv.org/pdf/2409.19681 SFD

---

> > ### Comment · Reviewer_xsUN · 2025-04-04
> >
> > I’ve read the authors’ response and appreciate their efforts. However, it seems they forgot to include the results supporting their claims, as several of the referenced tables are missing.

---

> > > ### Author Response · Authors · 2025-04-07
> > >
> > > Thank you so much for taking the time to read our rebuttal, and our sincere apologies for missing the attached tables in our first response, the links to which were accidentally removed trying to get under the 5000 character limit. We also apologize for the further delay; we spent an extra day to hopefully ensure that this response comprehensively answers your questions w/ more results, particularly if we can't continue our back-and-forth.
> > >
> > > ---
> > >
> > > ## S4S vs. Coeff. Distill. Methods
> > > **Rebuttal Tables 1-4**: https://imgur.com/a/2ngs7jR.
> > >
> > > Briefly, we find that S4S outperforms comparable methods for coefficient distillation across a large number of datasets. Oftentimes, at 7 NFEs, S4S achieves a similar performance that alternative methods reach using 10 NFEs.
> > >
> > > ## S4S-Alt vs. BNS
> > > **Rebuttal Tables 5-8:** https://imgur.com/a/ZN2kZG9
> > >
> > > We reimplemented BNS on our diff. models using the available information from the BNS paper. We also evaluated BNS with LPIPS as the distance metric for a more fair comparison + more extensive ablations. We find S4S-Alt outperforms BNS, particularly on low-NFE scales. A summary of these findings:
> > >
> > > - Our CIFAR results for BNS are similar to that of BNS; although there are discrepancies in the precise FID numbers, BNS trained their own CIFAR DM w/ different amounts of training time. The fact that this overall trend is similar, however, gives us reasonable confidence in our implementation.
> > > - Our results accentuate the importance of the alternating objective, as well as its robustness across choice of $r$ and distance metric, especially in Reb. Tables 6 and 8.
> > > - Using a fixed solver order (3) helps vs. using the maximum order allowed (as in BNS), especially with a larger number of NFEs. The exception to this is at 4 NFE, where having an extra parameter (order 4 solver) can give a small boost.
> > > - Relaxing the objective (i.e. $r>0$) can lead to very meaningful improvements in FID, whereas BNS always uses $r=0$.
> > > - Using PSNR as a distance metric is harmful at low-NFE scales. Reb. Tables 5 and 7 show that using PSNR reduces the FID of S4S-Alt, especially for CIFAR.
> > > - Using $r=0$ can be helpful for very high-order solvers like BNS with more (8+) NFEs. This can be seen in Reb. Tables 6 and 8, where BNS + LPIPS does better than S4S-Joint w/ max order at higher NFEs.
> > >
> > > This wholesale evaluation shows that while BNS is a strong algorithm for exploring the  solver design space, S4S-Alt gives overall better performance through our particular design choices.
> > >
> > > ## Trajectory Matching vs. Output Matching
> > >
> > > **Figure from Zhou et al. (Simple and Fast Distillation 2024)**: https://imgur.com/a/EgcOKEY
> > >
> > > **Rebuttal Tables 9-10**: https://imgur.com/a/JsVmems
> > >
> > > Given a disc. step sched., we trained S4S to match a teacher solver trajectory in two different ways:
> > >
> > > - Matching the teacher trajectory at uniform intervals along the time disc. (i.e. matching the trajectory at every 5th time step)
> > > - Matching the teacher trajectory on the GITS (https://arxiv.org/pdf/2405.11326) disc. schedule, which takes smaller steps where the avg. traj. deviation in the teacher is large (more curvature in traj.) and larger steps where deviation is smaller (less curv.).
> > >
> > > All other implementation details (i.e. $r$, LPIPS) are the same. Our results are in Reb. Tables 9-10. Briefly:
> > >
> > > - Matching the final output outperforms matching the trajectory, for 4, 5, and 10 NFEs.
> > > - Both traj. matching methods are reasonably good at "high"-NFE generation, but perform very poorly on few-NFE generation.
> > > - Uniform matching doesn't take into account the curv. of the teacher traj., requiring the student solver to learn to match difficult transitions between disc. steps.
> > > - GITS matching leads to much worse performance with few NFEs because the highest curvature areas (as displayed in Fig above) often have the most variation btwn. teacher traj. (aka pathological regions).
> > > - **Matching the teacher trajectory means one is unable to use LD3 for the student solver's time disc.**, since it no longer has the same time disc. as the teacher. All "matching" methods are outperformed by simply using iPNDM solver + LD3, much less using S4S as well.

---

### Official Review · Reviewer_qWi4 · 2025-03-11

**Overall Recommendation:** 5

**Summary:**

The paper proposes S4S and S4S-Alt, methods to optimize diffusion ODE solvers for fast, high-quality sampling with minimal neural function evaluations (NFEs). S4S learns solver coefficients via a distillation objective, matching the output of a high-NFE "teacher" solver while minimizing global error (not local truncation error). Additionally, S4S-Alt jointly optimizes solver coefficients and discretization steps via alternating minimization, further improving performance. The authors achieve state-of-the-art FID scores across datasets (e.g., 3.73 on CIFAR-10, 13.26 on MS-COCO with 5-8 NFEs), outperforming prior solvers like DPM-Solver++ and iPNDM. The method is lightweight (<1 A100 hour), data-free, and compatible with existing schedules.

**Claims And Evidence:**

Yes.

**Essential References Not Discussed:**

No.

**Experimental Designs Or Analyses:**

Strengths: Extensive evaluation across various datasets (CIFAR-10, FFHQ, ImageNet, etc.) and multiple baselines (DPM-Solver++, iPNDM, UniPC). Ablations validate design choices (time-dependent coefficients, relaxed objective).

**Methods And Evaluation Criteria:**

Yes.

**Other Comments Or Suggestions:**

No.

**Other Strengths And Weaknesses:**

Strengths: The proposed approaches are novel, and demonstrate universal improvement across datasets, architectures, and schedules. Moreover, they require no data/retraining, are computationally efficient, and can be plugged in black-box on top of any discretization schedule or architecture. The authors have performed extensive and impressive experiments to demonstrate the effectiveness of the proposed approaches.

I have not found major weaknesses in this submission.

**Questions For Authors:**

- How sensitive is performance to the radius $r$? Does the $r \propto m^{-5/2}$ heuristic (mentioned in Appendix G.2) hold across varying $m$?

- Can S4S/S4S-Alt achieve competitive performance with 1-2 NFEs, or does it require a minimum step count?

**Relation To Broader Scientific Literature:**

N/A

**Theoretical Claims:**

N/A. There are no theoretical results in the main document (and in the supplementary material, the authors only restate the theoretical guarantee for the relaxed objective presented in Eq. (7); this guarantee was provided by Tong et al. (2024)).

---

> ### Author Rebuttal · Authors · 2025-04-01
>
> Thank you for your time reviewing our work, and for recommending acceptance! We hope we answer your outstanding questions below.
>
> > How sensitive is performance to the radius $r$? Does the $r\propto m^{-5/2}$ heuristic hold across varying $m$?
>
> In practice, the heuristic $r \propto m^{-5/2}$ works reasonably well "out of the box" across the experimental settings we looked at. Sensitivity emerges in two different directions though on both sides of the parameter scale. With very few number of parameters, choosing the radius to be too small makes the optimization problem more difficult. On the other hand, with a larger number of parameters, allowing for too large of a radius enables more overfitting. While $m^{-5/2}$ is a good heuristic that achieves baseline strong performance in both settings, we would expect improvements to come by carefully tuning this parameter based on the experimental settings.
>
> To help clarify this, we ran some additional experiments on CIFAR-10 and ImageNet-256 to characterize this dependence; our results are in the table below.
>
> | Dataset | Model | Parameters $(m)$ | NFEs | $r=0.1m^{-5/2}$ | $r=0.5m^{-5/2}$ | $r=1.0m^{-5/2}$ | $r=2.0m^{-5/2}$ |
> |---------|-------|-----------------|------|-----------------|-----------------|-----------------|-----------------|
> | CIFAR-10 | iPNDM-S4S | 6 | 4 | 32.15 | 30.58 | 28.91 | 31.24 |
> | CIFAR-10 | S4S-Alt | 16 | 4 | 17.23 | 16.05 | 16.95 | 18.39 |
> | ImageNet | iPNDM-S4S | 6 | 4 | 8.12 | 7.84 | 8.06 | 8.53 |
> | ImageNet | S4S-Alt | 16 | 4 | 5.28 | 5.13 | 5.37 | 5.72 |
>
> As shown in the table, the performance is generally robust within a reasonable range around our heuristic, with the optimal value typically falling between $0.5m^{-5/2}$ and $1.0m^{-5/2}$. The sensitivity increases with larger model sizes, supporting our conclusion that a well-calibrated radius is more important as parameter count increases to prevent overfitting.
>
> > Can S4S/S4S-Alt achieve competitive performance with 1-2 NFEs, or does it require a minimum step count?
>
> Thank you for the interesting question! In practice, we found that going below 3 NFEs requires solving a very difficult problem that likely requires directly modifying the underlying score network. Concretely, 1 NFE generation entails directly generating an image from a noise latent; as such we completely lose any "degrees of freedom" over the time discretization; similarly, we may now only control ~2-4 coefficients, which is generally insufficient for producing high-quality outputs. While having 2 NFEs yields better performance, on traditional score network architectures, S4S and S4S-Alt still fall short of the performance seen in 1-2 step generation in training-based distillation methods. Nonetheless, this challenge persists for all other methods that don't modify the underlying score network.
>
> To further characterize these results, we conducted a few more experiments. First, for CIFAR-10, we characterized S4S and S4S-Alt on 2 NFE generation for LD3 discretization.
>
> | Dataset | Method | NFE=2 | NFE=3 | NFE=4 |
> |---------|--------|-------|--------|-------|
> | CIFAR-10 | iPNDM | 155.37 | 23.64 | 9.06 |
> | CIFAR-10 | iPNDM-S4S | 142.45 | 20.65 | 8.25 |
> | CIFAR-10 | S4S-Alt | 104.62 | 14.71 | 6.52 |
>
> As shown in the table, while S4S-Alt achieves significant improvements over traditional solvers at 2 NFEs, the FID scores are still substantially higher than at 3-4 NFEs. This supports our observation that there's a minimum effective step count (~3 NFEs) for maintaining reasonable image quality with training-free methods that don't modify the score network architecture.
>
> On the other hand, we decided to replicate an experiment from the LD3 paper that we didn't have time to include in our results. Here, we examined results on InstaFlow, a flow network that is explicitly trained for high-quality few-step sampling. When the underlying model is explicitly trained to produce high-quality few-step samples, S4S correspondingly improves the image quality as well, on a scale competitive with the teacher solver (8 NFEs, Uniform Time Disc. = 14.16 FID).
>
> | Method | NFE=2 |
> |---------|--------|
> | InstaFlow | 24.13 |
> | InstaFlow [LD3] | 16.74|
> | InstaFlow [LD3 + S4S] | 15.22 |
> | InstaFlow [LD3 + S4S-Alt] | 14.31 |

---

### Official Review · Reviewer_Ngqf · 2025-03-13

**Overall Recommendation:** 3

**Summary:**

The paper proposes the S4S method for optimizing diffusion model solvers. The optimization space includes the solver coefficients, time discretization schedule, and time correction terms. The optimization objective is a relaxed version of the global error with LPIPS as the distance metric, which only requires the existence of an input xT' sufficiently close to the original xT. Experiments demonstrate that S4S can outperform previous learning-free/learning-based solvers and fixed/learned timestep discretizations over diverse datasets.

**Claims And Evidence:**

Yes.

**Essential References Not Discussed:**

N/A

**Experimental Designs Or Analyses:**

Yes.

**Methods And Evaluation Criteria:**

Yes.

**Other Comments Or Suggestions:**

I suggest the authors visualize the learned coefficients and timestep discretizations in comparison to previous solvers.

**Other Strengths And Weaknesses:**

Strengths:
- The authors did a nice job in appendix A comparing to previous works, including (1) optimizing time steps (2) optimizing local truncation error (3) optimizing global error.
- Ablations are comprehensive, including (1) the benefits of alternating optimization (2) the benefits of relaxed objective (3) the disadvantage of large order, as in BNS (4) the benefits of LPIPS as distance metric (5) the initialization method (6) the training dataset size.

Weakness:
- With a larger parameter space, the method is more data-driven and less theory-grounded than traditional solvers. The coefficients require separate optimization under different NFE.
- There are no intuitive visualizations of the learned coefficients and timestep discretizations.

**Questions For Authors:**

- Does the sampling trajectory of the learned solver still follow the teacher's, or is only the final sampler closer?

**Relation To Broader Scientific Literature:**

The paper can optimize both the solver coefficients and timestep discretization with a novel alternating objective. The optimization cost and final performance are better than those of previous works.

**Theoretical Claims:**

The paper applies a practical objective to optimizing the solver in a free space. Unlike more restricted papers like DPM-Solver-v3, there are no strict theorems.

---

> ### Author Rebuttal · Authors · 2025-04-01
>
> Thank you for your time reviewing our work and for your recommendation of acceptance. Below, we hope to address the weaknesses you identified and the questions you raised.
>
> > With a larger parameter space, the method is more data-driven and less theory-grounded than traditional solvers. The coefficients require separate optimization under different NFE.
>
> We agree that our approach is less grounded in theory relative to other diffusion model solvers. However, we think that this is more a feature of the particular problem setting -- very low-NFE sampling -- than of our specific approach. For instance, in many impressive theory-grounded diffusion model solvers, e.g. DPM-Solver++, the underlying assumptions that guarantee convergence in these solvers begin to break down in the low-NFE regime (~3-5 steps) as the step size for each solver step increases. Accordingly, we see significant degredation in performance in many of these theory-grounded solvers as the number of NFEs decrease; this decrease in performance can be seen our tables in Appendix H. S4S's data-centric approach avoids trying to make these strong assumptions in the low-NFE regime and as a result achieves stronger performance than its theoretically-grounded alternatives. Nonetheless, from a theoretical perspective, it's still **not intuitive** why S4S even works; in fact, it seems shocking that solvers like iPNDM or DPM-Solver++ get remotely reasonable performance in the low-NFE regime in the first place. We think that trying to characterize why these approaches even achieve a modicum of success with few NFEs is a very exciting direction of future work that we hope to provide answers to.
>
> We also recognize that our data-centric approach necessitates S4S to learn a new solver for each number of NFEs. In practice, this is a similar limitation to alternative methods for learning diffusion model solvers (e.g. LD3 [1], BNS [2], $\Pi$A [3]). Creating a reusable method for crafting diffusion model sovlers without needing to retrain each time is similarly a direction for future work that we aim to address.
>
> > There are no intuitive visualizations of the learned coefficients and timestep discretizations.
>
> Thank you for raising this point! Here, we provide some visualizations of the learned time-step discretizations for [LSUN bedroom](https://imgur.com/a/uGAP88Q) and for [FFHQ](https://imgur.com/a/rFKizhf). These visualizations indicate that the learned time-step discretizations are similar to the "best" heuristic time discretizations, i.e. the learned time discretization for latent diffusion models are similar to the uniform time discretization, while the learned time discretizations are similar to EDM / logSNR for pixel-space models.
>
> > Does the sampling trajectory of the learned solver still follow the teacher's, or is only the final sampler closer?
>
> Thank you for asking this question! The answer to this question depends on the difficulty of the underlying task. Interestingly, in relatively simple domains (e.g. CIFAR-10), the trajectory of the learned solver still closely follows that of the teacher's, despite being trained to explicitly only match the output of the teacher solver. In contrast, however, on more complex domains (e.g. conditional generation in ImageNet-256 or MS-COCO text-to-image), the trajectories of the student solver can have notable differences from that of the teacher solver.

---

### Decision · Program_Chairs · 2025-05-01

**Decision:**

Accept (poster)

**Comment:**

This paper aims to optimize diffusion ODE solvers for high-quality sampling in the low-NFE regime. The method is lightweight, training-free, and compatible with existing schedules. All reviewers gave positive scores. They commended the paper for its clear presentation, a well-articulated discussion of its differences and novelty compared to previous works, and a comprehensive, promising, and persuasive experimental evaluation. The AC concurs, noting that the proposed method establishes a new benchmark for few-step sampling in diffusion models, holds great potential for wide-ranging applications in AIGC, and thus recommends acceptance.